# Glutamylation imbalance impairs the molecular architecture of the photoreceptor cilium

Olivier Mercey [ID][1], Sudarshan Gadadhar [ID][2,3,4], Maria M Magiera [ID][2,3], Laura Lebrun [ID][2,3], Corinne Kostic [ID][5], Alexandre Moulin[6], Yvan Arsenijevic [ID][7], Carsten Janke [ID][2,3✉], Paul Guichard [ID][1✉] & Virginie Hamel [ID][1✉]

## Abstract

Microtubules, composed of conserved α/β-tubulin dimers, undergo complex post-translational modifications (PTMs) that fine-tune their properties and interactions with other proteins. Cilia exhibit several tubulin PTMs, such as polyglutamylation, polyglycylation, detyrosination, and acetylation, with functions that are not fully understood. Mutations in AGBL5, which encodes the deglutamylating enzyme CCP5, have been linked to retinitis pigmentosa, suggesting that altered polyglutamylation may cause photoreceptor cell degeneration, though the underlying mechanisms are unclear. Using super-resolution ultrastructure expansion microscopy (U-ExM) in mouse and human photoreceptor cells, we observed that most tubulin PTMs accumulate at the connecting cilium that links outer and inner photoreceptor segments. Mouse models with increased glutamylation (Ccp5$^{-/-}$ and Ccp1$-/-$) or loss of tubulin acetylation (Atat1$-/-$) showed that aberrant glutamylation, but not acetylation loss, disrupts outer segment architecture. This disruption includes exacerbation of the connecting cilium, loss of the bulge region, and destabilization of the distal axoneme. Additionally, we found significant impairment in tubulin glycylation, as well as reduced levels of intraflagellar transport proteins and of retinitis pigmentosa-associated protein RPGR. Our findings indicate that proper glutamylation levels are crucial for maintaining the molecular architecture of the photoreceptor cilium.

**Keywords** Glutamylation; Post-translational Modifications; Photoreceptor Cell Cilium; Expansion Microscopy; Retinitis Pigmentosa
**Subject Categories** Cell Adhesion, Polarity & Cytoskeleton; Post-translational Modifications & Proteolysis

## Introduction

Cilia are highly conserved organelles present on the surface of most eukaryotic cells. They exhibit remarkable structural complexity organized around an axoneme composed of nine microtubule doublets. Tubulin, the constituent of these microtubules, undergoes a diverse array of PTMs including acetylation, detyrosination, polyglutamylation, polyglycylation, phosphorylation, polyamination, SUMOylation, glycosylation, arginylation, methylation or palmitoylation. These modifications, referred to as one aspect of the tubulin code, are thought to influence on microtubule dynamics, stability, and interactions with associated proteins (Janke and Magiera, 2020; Roll-mecak, 2020). PTMs being mostly enriched on stable microtubules, cilia represent an interesting model to study these modifications. Furthermore, tubulin PTMs have emerged as key regulators of ciliary assembly, maintenance, and signaling (Yang et al, 2021).

Polyglutamylation, one of the most abundant PTMs in cilia (Yang et al, 2021) is probably the most studied tubulin PTM in these organelles. In motile cilia, it has been shown that glutamylation controls the activity of inner arm dynein, important for the regulation of ciliary beating (Kubo et al, 2010; Suryavanshi et al, 2010). Recently, Alvarez Viar and colleagues revealed that polyglutamylation of protofilament 9 of the B-tubule is a conserved feature of motile cilia shared between algae and mice (Viar et al, 2023). This highly localized distribution of this PTM allows for the interaction with the nexin-dynein regulatory complex (NDRC), and thus regulating ciliary beating behavior. Interestingly, both hypoglutamylation (Grau et al, 2013; Ikegami et al, 2010; Pathak et al, 2011; Kubo et al, 2010; Suryavanshi et al, 2010) and hyperglutamylation (Pathak et al, 2014) impact ciliary motility in different model organisms, revealing that precisely controlled levels of glutamylation are crucial to maintain proper ciliary function. Several studies also showed that glutamylation regulates Intra-Flagellar Transport (IFT) dynamics, a bidirectional motility of ciliary constituents along axonemal microtubules required for assembly and maintenance (Scholey, 2003). Polyglutamylation positively regulates the IFT and certain microtubules motors

[1]Department of Molecular and Cellular Biology, University of Geneva, Geneva, Switzerland. [2]Institut Curie, PSL Research University, CNRS UMR3348, Orsay, France. [3]Université Paris-Saclay, CNRS UMR3348, Orsay, France. [4]Institute for Stem Cell Science and Regenerative Medicine (inStem), GKVK Post, Bellary Road, Bangalore, India. [5]Group for Retinal Disorder Research, Department of Ophthalmology, University Lausanne, Jules-Gonin Eye Hospital, Fondation Asile des Aveugles, Lausanne, Switzerland. [6]Department of Ophthalmology, University Lausanne, Jules-Gonin Eye Hospital, Fondation Asile des Aveugles, Lausanne, Switzerland. [7]Unit of Retinal Degeneration and Regeneration, Department of Ophthalmology, University Lausanne, Jules-Gonin Eye Hospital, Fondation Asile des Aveugles, Lausanne, Switzerland. ✉E-mail: Carsten.Janke@curie.fr; Paul.Guichard@unige.ch; Virginie.Hamel@unige.ch

(O'Hagan et al, 2011; Sirajuddin et al, 2014) whereas removal of polyglutamylation in cilia of engineered cell lines impairs anterograde IFT dynamics (Hong et al, 2018). Consequently, glutamylation is also important for ciliary signaling, as it impacts the localization of signaling molecules, such as Polycystins (He et al, 2018; O'Hagan et al, 2011), or Sonic Hedgehog components (Hong et al, 2018). Given the variety of ciliary functions associated with polyglutamylation, it is not surprising that mutations in enzymes generating or removing polyglutamylation are linked to ciliopathies.

Polyglutamylation is catalyzed by enzymes belonging to the tubulin-tyrosine ligase-like (TTLL) family (Janke et al, 2005; van Dijk et al, 2007). TTLL enzymes catalyze the addition of glutamate chains on tubulin and other substrates (Edde et al, 1990; van Dijk et al, 2008). These glutamate chains branch off the main peptide chain of the substrate, and their features appear to be controlled by the enzymatic specificities of the different TTLL enzymes: the preference for the generation of short versus long glutamate chains, as well as for modifying α- or β-tubulin (van Dijk et al, 2007). Polyglutamylation is reversible with deglutamylation catalyzed by enzymes from the cytosolic carboxy-peptidase (CCP) family (Rogowski et al, 2010; Tort et al, 2014), out of which some preferentially remove long glutamate chains, while others are more specific to short chains and the branching points. These enzymatic specificities are expected to translate into defined physiological functions, which is best illustrated by the phenotypes found in cilia and flagella when TTLL or CCP enzymes are selectively deleted. Indeed, a wide variety of ciliary defects have been linked to glutamylation imbalance. Deletion of polyglutamylases of the TTLL family lead to defects in cilia and flagella assembly and function. For instance, mutation of TTLL1 causes male infertility, massive defects in the assembly of the sperm flagellum, defects in the function of the nasal (Vogel et al, 2010), and airway ciliated epithelia (Ikegami et al, 2010). Mutations of TTLL5 cause retinal dystrophy in humans, presumably because of impaired glutamylation of the X-linked Retinitis Pigmentosa GTPase regulator RPGR (Sun et al, 2016). Mutations in the RPGR gene have been linked to retinitis pigmentosa, the most prevalent family of inherited retinal diseases leading to photoreceptor death. Concerning the CCP enzyme family, mutations of the deglutamylase CCP1 (Fernandez-Gonzalez et al, 2002), lead to a massive pathological increase in polyglutamylation in several organs and cell types (Rogowski et al, 2010). The mouse model has been known for decades as the Purkinje Cell Degeneration (pcd) mouse (Mullen et al, 1976). Pcd mice show - aside from the characteristic degeneration of cerebellar Purkinje cells - a variety of ciliary dysfunctions such as male infertility and photoreceptor degeneration (Mullen et al, 1976; LaVail et al, 1982). The causative role of abnormally increased polyglutamylation was demonstrated by knocking out the polyglutamylase TTLL1 in the pcd background, which entirely prevented degeneration of Purkinje cells and peripheral nerves (Magiera et al, 2018; Bodakuntla et al, 2021). The discovery of a novel early-onset neurodegeneration linked to inactivation of CCP1 underpinned the validity of the mechanisms discovered in mouse models for human health (Shashi et al, 2018).

Mutations in the *AGBL5*, the gene coding for the deglutamylase CCP5, cause defects in sperm development; CCP5-KO mice are unable to form functional flagella (Giordano et al, 2019). In humans, CCP5 mutations have also been associated with retinitis

pigmentosa (Astuti et al, 2016; Branham et al, 2016; Kastner et al, 2015). This provided the first link between tubulin polyglutamylation and photoreceptor degeneration in humans. Recently, a mouse model lacking CCP5 showed that hyperglutamylation observed in this mutant leads to photoreceptor cell degeneration with altered outer segments (Aljammal et al, 2024). However, mechanisms of photoreceptor cell degeneration at play are not known.

Here, we elucidate the role of tubulin PTMs in the outer segment (OS) of mouse photoreceptor cells. Using super-resolution Ultrastructure expansion microscopy (U-ExM) technique allowed us to obtain a deeper understanding of tubulin PTMs distribution inside the OS with nanoscale precision. Analyzing mutant mice deficient for key enzymes of tubulin PTMs, we unraveled that imbalance of glutamylation inside the OS results in a molecular and architectural disorganization of the photoreceptor axonemes, impairs photoreceptor function, and leads to retinal degeneration.

## Results

### The outer segment of mouse rod photoreceptor cells is enriched in tubulin PTMs

Using Ultrastructure Expansion Microscopy (U-ExM) that we recently adapted for retina imaging (Mercey et al, 2022; Gambarotto et al, 2018), we first assessed the nanoscale localization of tubulin PTMs along the rod photoreceptor cilium—the outer segment—in mouse retina with an expansion factor of 4.2× (Fig. EV1A,B). We focused on the longitudinal and transversal views comprising the basal body, the connecting cilium, where many retinopathy-associated proteins localize (Bachmann-Gagescu and Neuhauss, 2019), the bulge region that we recently described as a crucial compartment for membrane disc formation (Faber et al, 2023) and the distal axoneme. To map the distribution of different PTMs along the axoneme of the outer segment, we co-stained all microtubules using a mixture of anti α- and β-tubulin antibodies with a panoply of antibodies specific to tubulin PTMs: monoglycylation (TAP952), acetylation (acetylated tubulin), glutamylation- and polyglutamylation (GT335 and PolyE) and detyrosination (detyrosinated tubulin) (Figs. 1A–E and EV1C).

We first mapped tubulin glycylation. This modification has been mainly found in motile cilia/flagella (Gadadhar et al, 2021; Grau et al, 2013) but is also present during primary cilia assembly (Rocha et al, 2014; Gadadhar et al, 2017). In the photoreceptor cilium, that can be considered as a specialized primary cilium, we found that glycylation is restricted to the connecting cilium and the bulge region, where it exactly lines microtubules (−0.6 nm shift relative to α- and β-tubulin staining) (Fig. 1A,F). In contrast, glycylation was mostly absent distally to the bulge and we confirmed that it was also absent from the basal body (Guichard et al, 2023). However, by oversaturating the signal, we found a reproducible faint signal that seems to localize at the level of subdistal appendages of the centriole (Appendix Fig. S1).

We next analyzed tubulin acetylation status in the outer segment. This modification takes place in the lumen of the microtubules (Fig. EV1C), acting on their mechanical properties (Eshun-Wilson et al, 2019). We found that acetylation is present from the basal body to the bulge region of the axonemal microtubules (Fig. 1B,F). However, similar to glycylation, the

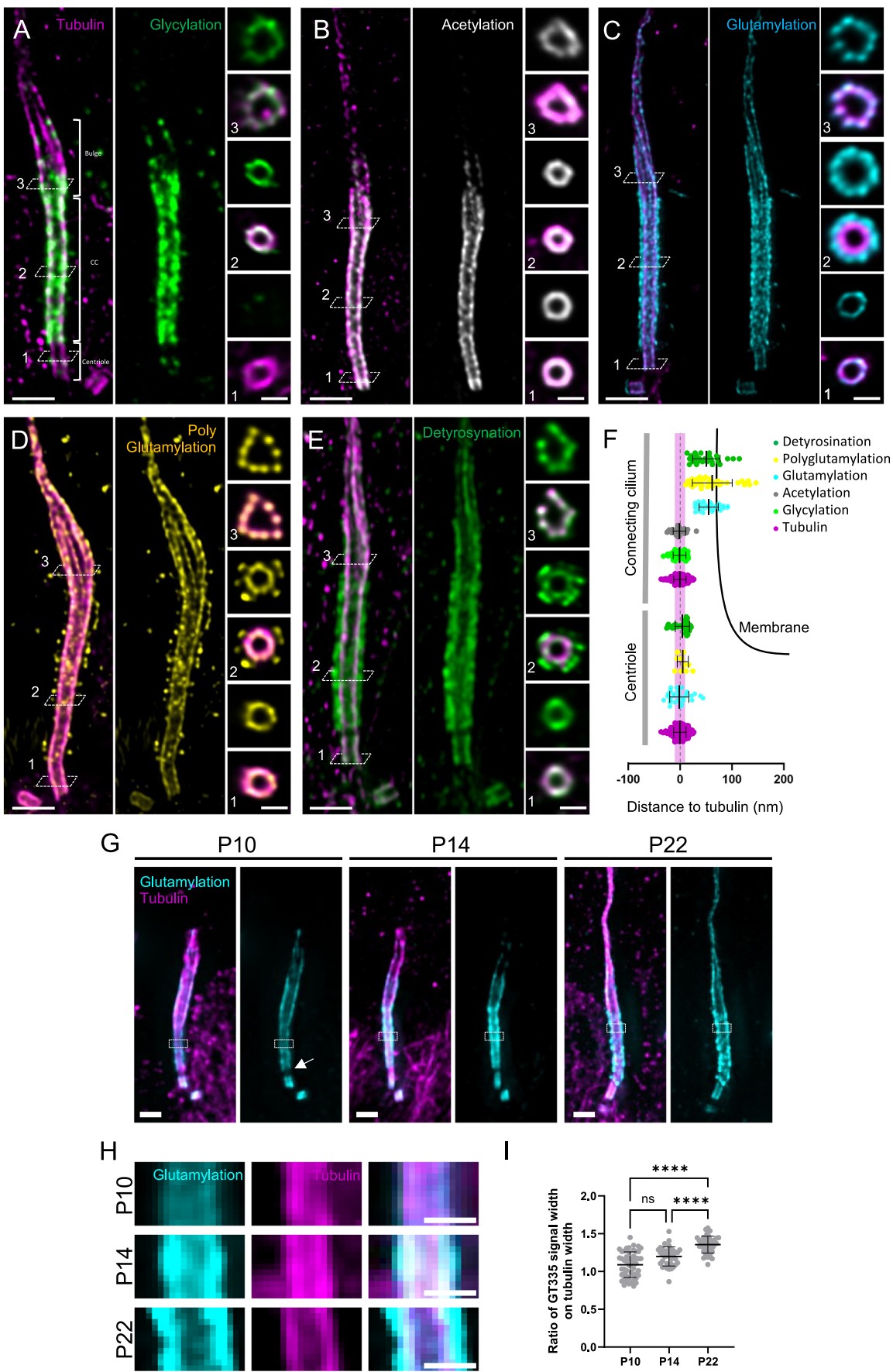

**Figure 1. Molecular mapping of tubulin PTMs in mouse photoreceptor cells.**

(A–E) Confocal images of expanded mouse photoreceptor cells stained with tubulin and (A) glycylation (TAP952, green), (B) acetylation Tubulin (gray), (C) glutamylation (GT335, cyan), (D) polyglutamylation (PolyE, yellow) or (E) detyrosination (green). Scale bar: 500 nm. Transversal section images corresponding to different regions of the OS (centriole, connecting cilium and bulge depicted by the dashed lines and numbers on longitudinal images) are represented on the right side. Scale bar: 200 nm. (F) Schematic representation of the outer segment centriole and connecting cilium on which measured distances to tubulin of different proteins are represented. Membrane is depicted as a black line. Width of a microtubule (20 nm) is depicted in magenta as a reference. Glutamylation (centriole): $-1.21$ nm $+/-$ 18.61 ($n = 33$; $N = 3$ animals); polyglutamylation (centriole): 5.66 nm $+/-$ 10.91 ($n = 10$; $N = 1$ animal); detyrosination (centriole): 4.92 nm $+/-$ 14.07 ($n = 33$; $N = 2$ animals); glycylation (CC): $-0.57$ nm $+/-$ 12.45 ($n = 50$; $N = 3$ animals); acetylation (CC): $-0.93$ nm $+/-$ 12.30 ($n = 37$; $N = 2$ animals); glutamylation (CC): 56.42 nm $+/-$ 18.90 ($n = 47$; $N = 3$ animals); polyglutamylation (CC): 63.5 nm $+/-$ 39.01 ($n = 35$; $N = 3$ animals); detyrosination (CC): 51.98 nm $+/-$ 26.94 ($n = 27$; $N = 2$ animals) (mean $+/-$ SD). Tubulin is used as a reference: 0 nm $+/-$ 12 ($n = 304$; $N > 10$ animals). (G) Developing photoreceptor OS at P10, P14 and P22 stained for glutamylation (GT335, cyan) and tubulin (magenta). Insets represent the regions where GT335 and tubulin signal width were highlighted in (H) and quantified in (I). White arrow indicates a gap of glutamylation between the centriole and the CC. Scale bar: 500 nm. (H) zoom in images from (G) highlighting the increase of glutamylation width at the CC between P10 and P22. Scale bar: 200 nm. (I) Quantification of GT335 signal width during OS development (P10 to P22) and normalized to tubulin width. P10: 1.09 $+/-$ 0.17 ($n = 51$; $N = 2$ animals); P14: 1.20 $+/-$ 0.13 ($n = 37$; $N = 2$ animals); 1.36 $+/-$ 0.11 ($n = 43$; $N = 2$ animals) (mean $+/-$ SD). Test: Kruskal–Wallis with Dunn's multiple comparison. P10 vs. P14: ns (adjusted $P$ value: 0.0535); P14 vs. P22: ****(adjusted $P$ value: <0.0001). P10 vs. P22: ****(adjusted $P$ value: <0.0001). Each animal corresponds to one experimental replicate. Source data are available online for this figure.

signal above the bulge was mostly absent or faint, suggesting that distal disorganized axonemal microtubules are less prone to be modified by glycylating or acetylating enzymes.

Next, we examined glutamylation localization with two different antibodies: GT335 raised against a synthetic peptide mimicking the glutamylation modification on a C-terminal tubulin tail; and PolyE, an antibody recognizing long glutamate side chains (>3 glutamates) (Shang et al, 2002; Rogowski et al, 2010) (Fig. 1C,D). For both antibodies, glutamylation staining reveals two localizations along the photoreceptor outer segment. The basal body is decorated with both glutamylation and polyglutamylation with the exact same localization as the tubulin, as previously shown in *Chlamydomonas* and human centrioles (Mahecic et al, 2020; Gambarotto et al, 2019) (Fig. 1C,D,F). However, the pattern changes drastically once the connecting cilium (CC) starts, with intense glutamylation (GT335 antibody) and polyglutamylation (polyE) signals more external to the microtubule wall and all along the CC, forming a sheath-like structure. We quantified this signal about 60 nm away from the microtubule center of mass (Fig. 1C,D,F), rather close to the membrane, where CEP290 has been recently localized (Mercey et al, 2022) (Fig. 1F). We also quantified the GT335 signal intensity at different locations and confirmed that the CC-associated GT335 signal is more intense compared to that of the centriole or the bulge (Fig. EV1D–F). The discrepancy between the position of tubulin and that of glutamylation suggests that this modification might decorate another protein(s) than tubulin, which are located closer to the membrane. One possibility is that we are also detecting the glutamylation of RPGR as previously observed (Sun et al, 2016) and also localized in this region (Takahashi et al, 2024). This external glutamylation signal is restricted to the connecting cilium, whereas the bulge and the distal axoneme exhibit glutamylation signal at the level of the microtubules, similar to the centriole. Moreover, unlike acetylation or glycylation signals, polyglutamylation seems to propagate more distally in the outer segment axoneme. To get a better impression of the precise localization, we also looked at the transverse view, where we confirmed the presence of a sheath-like glutamylation signal at the connecting cilium (Fig. 1C,D). Additionally, polyglutamylation staining was observed on the tubulin, a signal not detected by the GT335 glutamylation antibody (Fig. 1D). We also noticed that in some cases, PolyE accumulates at the base of the cilium, a signal that resembles IFT train accumulation in U-ExM (Appendix Fig. S2, white arrows)

(Van Den Hoek et al, 2022). To further elucidate glutamylation levels in the photoreceptor cilium, we analyzed the GT335 signal at several early stages of outer segment (OS) development, specifically at P10, P14, and P22 (Fig. 1G–I). At P10, the GT335 signal was notably intense at centrioles and was already detectable along the developing OS microtubule axoneme. However, the signal was not continuous from the centriole to the distal cilium, as evidenced by a gap between the centriole and the connecting cilium (CC) (Fig. 1G, white arrow). Interestingly, at P10, the GT335 signal closely overlapped with the tubulin signal in width, but as development progressed to P14 and P22, the GT335 signal became wider and more intense. This suggests that the sheath-like localization observed is not initially present at early timepoints but instead emerges later during OS development (Fig. 1H,I).

Finally, we also assessed the profiles of detyrosinated tubulin and Δ2-tubulin. These two tubulin PTMs consist in the removal of the gene-encoded C-terminal tyrosine (detyr-tubulin), and the further cleavage of the penultimate glutamate residue (Δ2-tubulin) of α-tubulin (Fig. EV1C). The staining pattern of detyrosinated tubulin closely mirrored that of glutamylation, and in particular for polyglutamylation as shown in the transverse view at the level of the CC (Fig. 1E). Detyrosinated tubulin is thus localized on microtubules, but also forms a sheath further away from them. Staining for Δ2-tubulin, by contrast, yielded a signal that was not distinct enough to conclude about the precise localization of this PTM (Appendix Fig. S3).

Altogether, these data reveal a complex distribution of tubulin PTMs along the photoreceptor outer segment. We found a specific enrichment at the level of the CC, where all PTMs analyzed are present, hinting at a prominent role of these modifications in this compartment.

## Molecular mapping of the tubulin PTMs in human photoreceptor cell outer segment

Next, we wanted to assess whether the observed distribution of tubulin PTMs is conserved in human retina, as mutations of *AGBL5*, coding for CCP5, lead to retinitis pigmentosa in human (Fig. 2A–F). To do so, we expanded human retina from a healthy adult. We first noticed that the length of the CC is shorter in human photoreceptor cells as compared to mouse. Using either GT335 or POC5 (Mercey et al, 2022) antibodies as markers of the

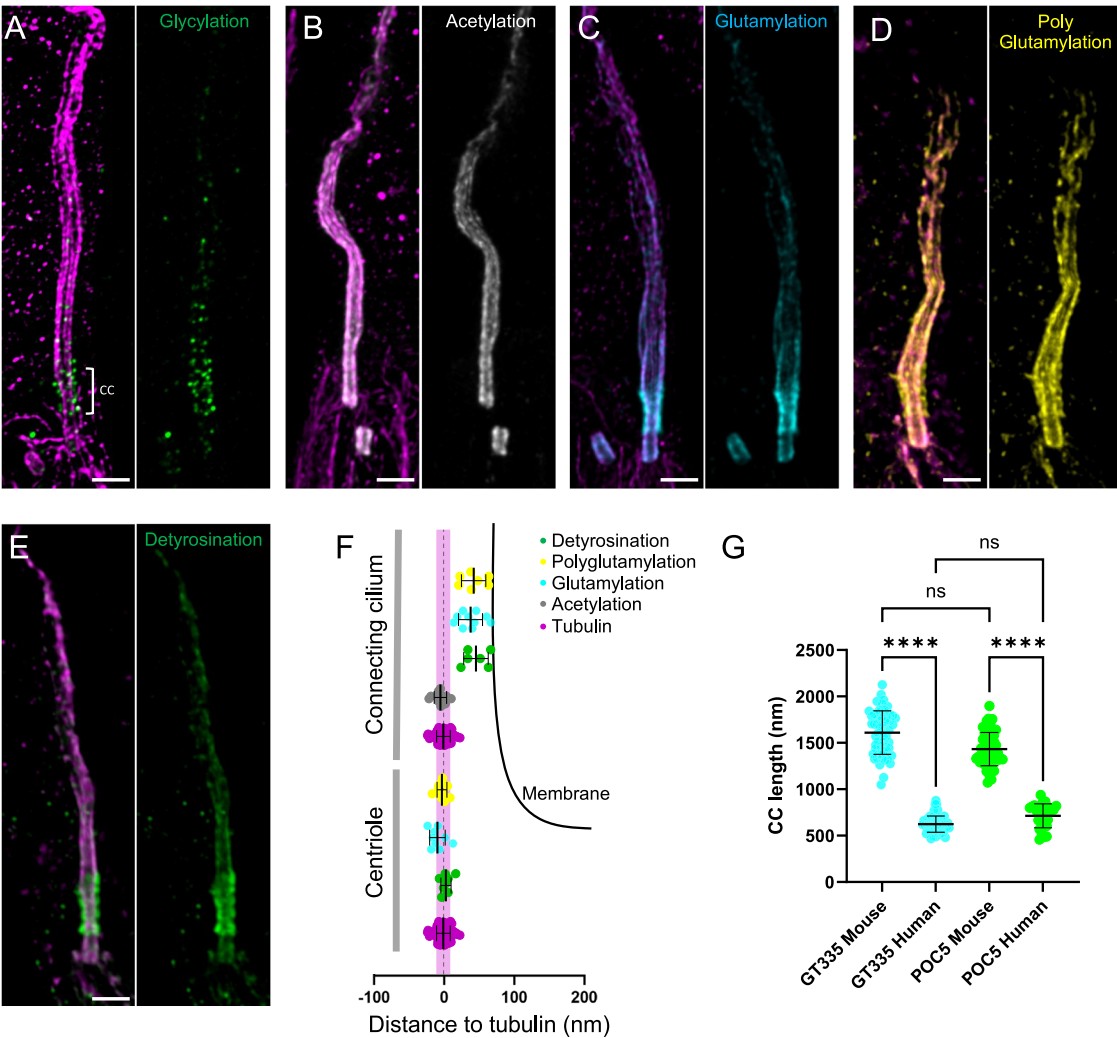

**Figure 2. Molecular mapping of tubulin PTMs in human photoreceptor cells.**

(A–E) Confocal images of expanded photoreceptor cells stained for tubulin (magenta) and (A) glycylation (TAP952, green), (B) acetylation (gray), (C) glutamylation (GT335, cyan), (D) polyglutamylation (PolyE, yellow) or (E) detyrosination tubulin (green). Scale bar: 500 nm. (F) Schematic representation of the outer segment centriole and connecting cilium on which measured distances to tubulin of different proteins are represented. Detyrosination (centriole): 3.73 nm +/− 7.29 ($n = 8$); glutamylation (centriole): −8.60 nm +/− 11.16 ($n = 9$); polyglutamylation (centriole): −2.03 nm +/− 7.12 ($n = 10$); acetylation (CC): −4.41 nm +/− 9.03 ($n = 12$); detyrosination (CC): 47.15 nm +/− 17.70 ($n = 6$); glutamylation (CC): 39.25 nm +/− 17.44 ($n = 10$); polyglutamylation (CC): 43.78 nm +/− 17.68 ($n = 7$) (mean +/− SD). Tubulin is used as a reference: 0 nm +/− 9.76 ($n = 62$) (mean +/− SD). Membrane is depicted as a black line. Width of a microtubule (20 nm) is depicted in magenta as a reference. (G) Comparison of mouse and human CC length using GT335 and POC5 as markers. GT335 (mouse): 1610 nm +/− 235 ($n = 57$; $N = 3$ animals); GT335 (human): 623 nm +/− 87 ($n = 60$); POC5 (mouse): 1433 nm +/− 179 ($n = 55$; $N = 3$ animals); POC5 (human): 714 nm +/− 130 ($n = 28$) (mean +/− SD). Test: Kruskal–Wallis with Dunn's multiple comparison. GT335 mouse vs. GT335 human: ****(adjusted $P$ value: <0.0001); GT335 mouse vs. POC5 mouse: ns (adjusted $P$ value: 0.0810); POC5 mouse vs. POC5 human: ****(adjusted $P$ value: <0.0001); GT335 human vs. POC5 human: ns (adjusted $P$ value: 0.6299). Human sample replicate: $N = 1$. Each animal corresponds to one experimental replicate. Source data are available online for this figure.

CC, we demonstrated that the human CC is less than half the length of the one in mouse, spanning about 650 nm vs ~1600 nm in mouse (Fig. 2G; Appendix Fig. S4). We also noticed that cone photoreceptors exhibit particularly long daughter centrioles that are more than 700 nm long, which could represent one of the longest centrioles observed in the human body (Appendix Fig. S4).

We show that glycylation signal (TAP952), while being faint, is present along the human CC and probably on the bulge, similarly to what we described in mouse (Fig. 2A). However, the signal was not strong enough to allow quantification. We also found that acetylation is decorating microtubules all along the OS, from the

basal body to the bulge, but also distally, which was less obvious in mouse (Fig. 2B,F). Next, analyzing glutamylation (GT335 and PolyE) and detyrosination signals, we observed the same pattern as in mouse, forming a sheath outside of the microtubules at the level of the CC, and lining the microtubules at the basal body and the cilium above the CC (Fig. 2C–E). The measured diameters revealed the same range of distances to microtubules as in mouse (Figs. 2F and 1F), highlighting the conservation of PTM localization along the photoreceptor outer segment in mouse and human. In line with this, a recent paper using U-ExM in canine photoreceptor cells described a similar sheath localization at the level of the CC using

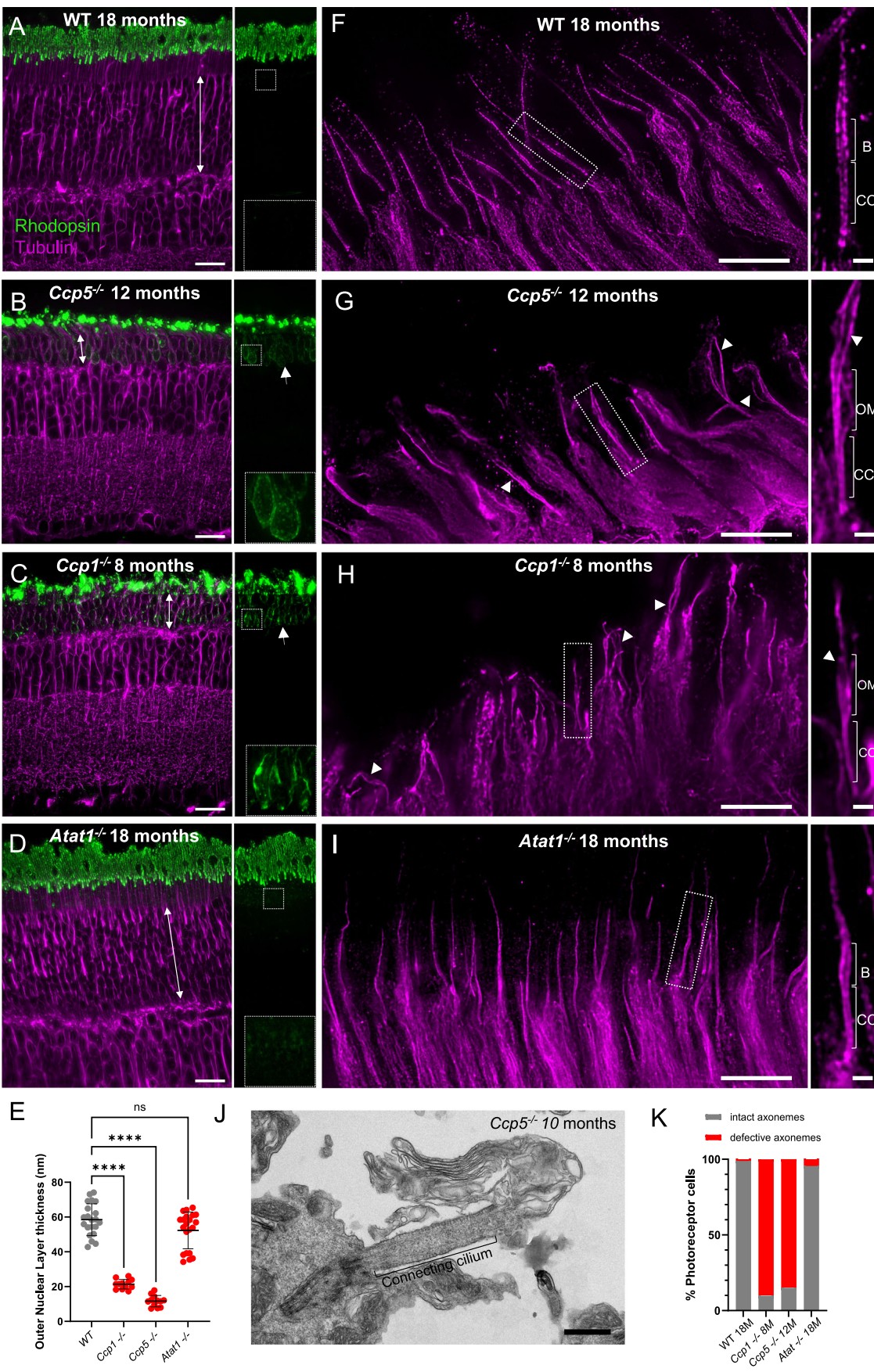

**Figure 3. Deglutamylase mutants lead to retinal degeneration with photoreceptor axonemal disorganization.**

(A–D) Expanded retina at low magnification stained for tubulin (magenta) and rhodopsin (green) in WT (A), *Ccp5*$^{-/-}$ (B), *Ccp1*$^{-/-}$ (C) or *Atat1*$^{-/-}$ (D) mice. On the right, the rhodopsin channel alone highlights the presence of the mislocalized signal in the ONL in the degenerating retina (white arrows and insets). The thickness of the Outer Nuclear Layer (ONL) is depicted with white double arrows. Scale bar: 20 μm. (E) Measurements of the ONL thickness in the different conditions, corrected by the expansion factor. WT: 58.47 nm +/− 9.15 ($n = 21$; $N = 3$ animals); *Ccp1*$^{-/-}$: 21.36 nm +/− 2.76 ($n = 12$; $N = 2$ animals); *Ccp5*$^{-/-}$: 11.65 nm +/− 3.27 ($n = 12$; $N = 2$ animals); *Atat1*$^{-/-}$: 52.31 nm +/− 10.54 ($n = 21$; $N = 3$ animals) (mean +/− SD). Test: Kruskal–Wallis with Dunn's multiple comparison. WT vs. *Atat1*$^{-/-}$: ns (adjusted P value: 0.9684); WT vs. *Ccp1*$^{-/-}$: ****(adjusted P value: <0.0001); WT vs. *Ccp5*$^{-/-}$: ****(adjusted P value: <0.0001). (F–I) Expanded photoreceptor cells stained with tubulin (magenta) highlighting the axonemal structure in WT (F), *Ccp5*$^{-/-}$ (G), *Ccp1*$^{-/-}$ (H) or *Atat1*$^{-/-}$ (I). Scale bar: 5 μm. Insets depicted with dashed lines are represented on the right. Scale bar: 500 nm. (J) EM micrograph of a 10-month-old *Ccp5*$^{-/-}$ photoreceptor outer segment revealing mostly intact CC, whereas the distal part of the cilium is highly impaired. Scale bar: 500 nm. (K) Percentage of individual photoreceptor cells with normal or abnormal axonemal structures. WT: normal: 98.89%, abnormal: 1.11% ($n = 180$; $N = 6$ animals); *Ccp5*$^{-/-}$: normal: 9.84%, abnormal: 90.16% ($n = 53$; $N = 2$ animals); *Ccp1*$^{-/-}$: normal: 15.1%, abnormal: 84.9% ($n = 61$; $N = 2$ animals); *Atat1*$^{-/-}$: normal: 95.5%, abnormal: 4.5% ($n = 89$; $N = 2$ animals) CC: Connecting Cilium; B: Bulge; OM: Open microtubules. White arrowheads show open microtubules in low magnification images. Each animal corresponds to one experimental replicate. Source data are available online for this figure.

GT335 antibody (Takahashi et al, 2024). In summary, U-ExM unveiled the conservation of PTM distribution along the photoreceptor cell OS but also highlighted structural organization differences between mouse and human photoreceptors cells.

## Glutamylation defects lead to defective axoneme architecture

Next, we investigated the physiological relevance of tubulin PTMs for photoreceptor cells. Indeed, as mutation in *AGBL5*, coding the deglutamylase CCP5, leads to retinitis pigmentosa in humans, it suggests that perturbation of glutamylation could lead to drastic consequences on photoreceptor survival. We thus analyzed retina of mice lacking CCP5 (*Ccp5*$^{-/-}$) and CCP1 (*Ccp1*$^{-/-}$). Absence of these two deglutamylating enzymes is expected to lead to hyperglutamylation (Rogowski et al, 2010). To test whether other PTMs than glutamylation could affect photoreceptor cells, we also assessed the consequence of loss of tubulin acetylation in *Atat1*$^{-/-}$ mice.

After expansion, we assessed the integrity of the retina in mice aged from 8 to 18 months old using rhodopsin, a marker of the rod outer segment, as well as α- and β-tubulin staining, in comparison to control mice (Fig. 3A–D). Knockout of the deglutamylases CCP1 or CCP5 lead to a severe photoreceptor degeneration at 8 and 12 months, respectively (Fig. 3B,C). This is highlighted by the substantial decrease of ONL thickness compared to wild type (WT), where only a couple of nuclear rows are remaining (Fig. 3B,C,E). This phenotype was already described in *pcd* (purkinje cell degeneration) mice, bearing an inactivating mutation in the *AGTPBP1* gene (coding for CCP1) (LaVail et al, 1982). At the level of the OS, rhodopsin is also highly impaired compared to the rod-like signal in the WT, suggesting disorganization of membrane discs. We confirmed this result in 7-month-old *Ccp5*$^{-/-}$ using Electron Microscopy (EM), where we observed disorganized membrane discs, and a reduction of cell number to about half compared to WT (Fig. EV2B). Additionally, we show a mislocalization of rhodopsin in *Ccp5*$^{-/-}$ and *Ccp1*$^{-/-}$, with a signal at the ONL-surrounding nuclei; a hallmark of photoreceptor degeneration (Fig. 3B,C, white arrows and insets). Interestingly, we noticed during dissections that retinas from *Ccp5*$^{-/-}$ mice are thinner and more fragile compared to WT (Fig. EV2C). Of note, the other layers of the retina were not affected in these mutants. By contrast, we show that ATAT1 deficiency has no overall impact on photoreceptor survival, even in 18-month-old mice (Fig. 3D).

Indeed, the thickness of the ONL is similar to control, with an intact rhodopsin staining (Fig. 3D,E).

We next focused the analysis on the photoreceptor outer segment, where PTMs are generally enriched (Fig. 3F–I). Given that CCP5 deficiency is linked to retinitis pigmentosa, and that we recently showed that FAM161A-associated retinitis pigmentosa RP28 is due to structural defects at the level of the CC (Mercey et al, 2022), we investigated whether this structural element was affected in deglutamylase mutants.

Using tubulin staining to reveal the architecture of the ciliary axoneme, we showed that WT photoreceptor OS have straight axonemal microtubule extending toward the distal part of the cilium (Fig. 3F). We also confirmed the presence of the bulge region, delineating the end of the CC (Figs. 3F and EV1A). By contrast, in the two deglutamylase mutants analyzed, the structure of the OS is impaired, mostly with open or disorganized axoneme on its distal end (Fig. 3G,H, white arrowheads). Interestingly, we found that for *Ccp5*$^{-/-}$ and *Ccp1*$^{-/-}$ mice, the structural defects were observed mostly above the CC, whereas the tubulin shaft seemed preserved at the CC, as it has been described in mice mutant for the bulge protein LCA5 (Faber et al, 2023) (Fig. 3G,H). Using EM in 10-month-old *Ccp5*$^{-/-}$, we confirmed that despite a severe membrane disc disorganization in the OS, CC seems mostly preserved (Fig. 3J). It should be also noted that more than 80% of the photoreceptor cells analyzed in *Ccp5*$^{-/-}$ and *Ccp1*$^{-/-}$ mutants are defective, highlighting the high penetrance of the degenerative phenotype in these mutants (Fig. 3K). By comparison, *Atat1*$^{-/-}$ photoreceptor cells have no overall axonemal defects in old mice (Fig. 3F,I), demonstrating that lack of tubulin acetylation does not result in obvious alteration at the level of photoreceptor cells.

## Deglutamylase deficiency disrupts distal axoneme organization and ciliary transport in the outer segment

To gain mechanistic insights into photoreceptor degeneration linked to PTMs and in particular hyperglutamylation, we next assessed the molecular architecture of photoreceptors by U-ExM, with a specific focus on the outer segment (OS), in different PTMs mutants. We analyzed staining for different PTMs (glycylation, glutamylation and acetylation), the CC marker POC5 (Mercey et al, 2022), the bulge marker LCA5 (Faber et al, 2023), and the Intraflagellar Transport (IFT) component IFT88.

Since lack of both CCP1 and CCP5 affect the glutamylation status, we first examined the GT335 pattern along the

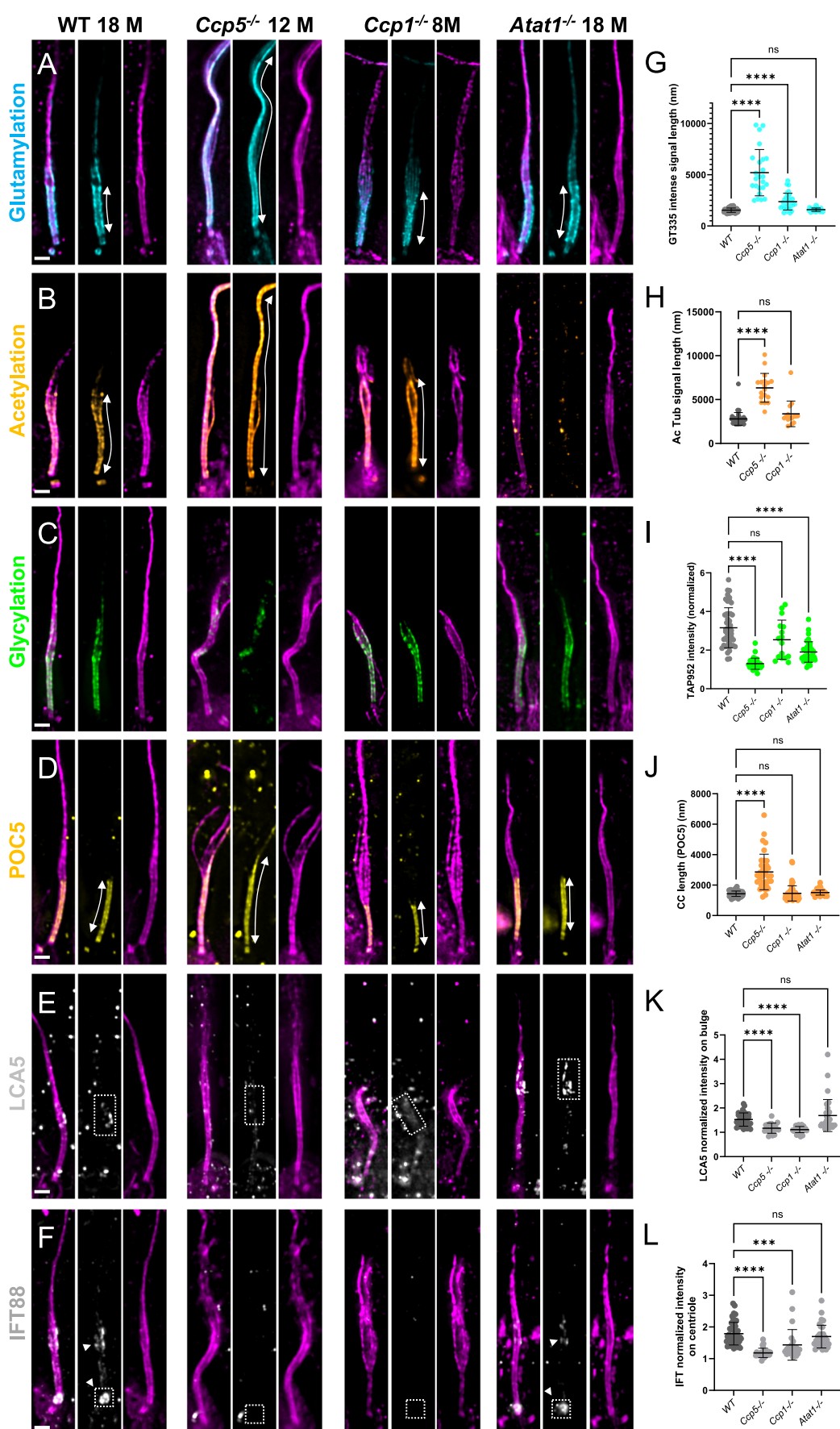

**Figure 4. Deglutamylase mutants cause imbalance in PTMs paralleled with defects in outer segment structure and transport.**

(A–F) Expanded photoreceptor outer segments observed in WT, $Ccp5^{-/-}$, $Ccp1^{-/-}$ or $Atat1^{-/-}$ and stained with GT335 (cyan) (**A**), Acetylated tubulin (orange) (**B**), TAP952 (green) (**C**), POC5 (yellow) (**D**), LCA5 (gray) (**E**), and IFT88 (gray) (**F**). White double arrows depict the length of the signal measured in (**G, H, J**). Rectangles with dashed lines show the area where intensity measurements were picked in (**K**) and (**L**). White arrowheads show the dual localization of IFT at the bulge and at the cilium entry. Scale bar: 500 nm. (**G**) GT335 intense signal length corrected by the EF. WT: 1546 nm $+/-$ 212.8 ($n = 30$; $N = 2$ animals); $Ccp5^{-/-}$: 5209 nm $+/-$ 2260 ($n = 24$; $N = 2$ animals); $Ccp1^{-/-}$: 2378 nm $+/-$ 814.2 ($n = 31$; $N = 2$ animals); $Atat1^{-/-}$: 1594 nm $+/-$ 166.1 ($n = 37$; $N = 2$ animals), (mean $+/-$SD). Test: Kruskal–Wallis with Dunn's multiple comparison. WT vs. $Ccp5^{-/-}$: ****(adjusted P value: <0.0001); WT vs. $Ccp1^{-/-}$: ****(adjusted P value: <0.0001); WT vs. $Atat1^{-/-}$: ns (Adjusted P value: >0.9999). (**H**). Acetylated tubulin signal length in the outer segment corrected by the EF. WT: 2795 nm $+/-$ 729.4 ($n = 43$; $N = 2$ animals); $Ccp5^{-/-}$: 6337 nm $+/-$ 1645 ($n = 18$; $N = 2$ animals); $Ccp1^{-/-}$: 3365 nm $+/-$ 1463 ($n = 15$; $N = 1$ animal), (mean $+/-$SD). Test: Kruskal–Wallis with Dunn's multiple comparison. WT vs. $Ccp5^{-/-}$: ****(adjusted P value: <0.0001); WT vs. $Ccp1^{-/-}$: ns (adjusted P value: 0.1085). (**I**) Intensity measurement of TAP952 signal along the outer segment CC normalized on background. WT: 3.16 nm $+/-$ 1.03 ($n = 47$; $N = 3$ animals); $Ccp5^{-/-}$: 1.30 $+/-$ 0.29 ($n = 34$; $N = 3$ animals); $Ccp1^{-/-}$: 2.54 $+/-$ 1.01 ($n = 16$; 2 animals); $Atat1^{-/-}$: 1.90 $+/-$ 0.53 ($n = 37$; 2 animals), (mean $+/-$SD). Test: Kruskal–Wallis with Dunn's multiple comparison. WT vs. $Ccp5^{-/-}$: ****(adjusted P value: <0.0001); WT vs. $Ccp1^{-/-}$: ns (adjusted P value: 0.2646); WT vs. $Atat1^{-/-}$: ****(adjusted P value: <0.0001). (**J**) CC length measured with POC5 and corrected by the EF. WT: 1433 nm $+/-$ 179 ($n = 55$; $N = 3$ animals); $Ccp5^{-/-}$: 2864 nm $+/-$ 1164 ($n = 36$; $N = 2$ animals); $Ccp1^{-/-}$: 1458 nm $+/-$ 498 ($n = 59$; $N = 2$ animals); $Atat1^{-/-}$: 1512 nm $+/-$ 167 ($n = 56$; $N = 3$ animals), (mean $+/-$SD). Test: Kruskal–Wallis with Dunn's multiple comparison. WT vs. $Ccp5^{-/-}$: ****(adjusted P value: <0.0001); WT vs. $Ccp1^{-/-}$: ns (adjusted P value: 0.5965); WT vs. $Atat1^{-/-}$: ns (Adjusted P value: 0.2385). WT measurements are identical to Fig. 2G. (**K**) LCA5 intensity at the bulge normalized on background. WT: 1.53 $+/-$ 0.27 ($n = 35$; $N = 3$ animals); $Ccp5^{-/-}$: 1.17 $+/-$ 0.21 ($n = 20$; $N = 2$ animals); $Ccp1^{-/-}$: 1.11 $+/-$ 0.13 ($n = 31$; $N = 2$ animals); $Atat1^{-/-}$: 1.69 $+/-$ 0.66 ($n = 33$; $N = 3$ animals), (mean $+/-$SD). Test: Kruskal–Wallis with Dunn's multiple comparison. WT vs. $Ccp5^{-/-}$: ****(adjusted P value: <0.0001); WT vs. $Ccp1^{-/-}$: ****(adjusted P value: <0.0001); WT vs. $Atat1^{-/-}$: ns (adjusted P value: >0.9999). (**L**) IFT88 intensity at the cilium base normalized on background. WT: 1.79 $+/-$ 0.36 ($n = 46$; $N = 3$ animals); $Ccp5^{-/-}$: 1.18 $+/-$ 0.15 ($n = 25$; $N = 2$ animals); $Ccp1^{-/-}$: 1.44 $+/-$ 0.48 ($n = 25$; $N = 2$ animals); $Atat1^{-/-}$: 1.70 $+/-$ 0.36 ($n = 32$; $N = 2$ animals), (mean $+/-$ SD). Test: Kruskal–Wallis with Dunn's multiple comparison. WT vs. $Ccp5^{-/-}$: ****(adjusted P value: <0.0001); WT vs. $Ccp1^{-/-}$: ****(adjusted P value: <0.0001); WT vs. $Atat1^{-/-}$: ns (adjusted P value: 0.5772). Each animal corresponds to one experimental replicate. Source data are available online for this figure.

photoreceptor OS. As expected, we observed a strong hyperglutamylation in $Ccp5^{-/-}$ where the signal extends towards the distal part of the cilium, thus decorating the whole axoneme (Fig. 4A,G). Remarkably, a similar pattern was observed for acetylated tubulin, where the signal was prolonged distally (Fig. 4B,H). Low magnification images revealed that hyperglutamylation is not restricted to the OS but extends to the inner segment of the photoreceptor cells in $Ccp5^{-/-}$ (Fig. EV3). Of note, in $Ccp1^{-/-}$, axonemes appear shorter than in $Ccp5^{-/-}$, probably explaining why hyperglutamylation and hyperacetylation towards the distal axoneme were less pronounced in these mutants. We also noticed that $Ccp5^{-/-}$, but not $Ccp1^{-/-}$ display a strongly reduced glycylation signal along the axoneme, underpinning that hyperglutamylation caused by the loss of some of the deglutamylases could lead to reduced glycylation, as previously described (Grau et al, 2017) (Fig. 4C,I).

In both $Ccp1^{-/-}$ and $Ccp5^{-/-}$, we confirmed a massive disorganization of the microtubules above the CC, with axoneme opening (Fig. 4A–F). Inside the CC, POC5 signal remains unaffected, suggesting that the CC inner scaffold is still present to maintain the cohesion of the microtubules in this compartment (Mercey et al, 2022), and that the observed photoreceptor cell degeneration is not due to structural defects of the CC. However, $Ccp5^{-/-}$ photoreceptor cells exhibit an exacerbated CC, with POC5 signal extending to more than 5 µm in some cases (compared to 1.5 µm in the control mice) (Fig. 4D,J). In contrast, CCP1 deficiency did not affect the length of the POC5 signal.

Next, we analyzed the bulge region, by investigating the distribution of LCA5 (Faber et al, 2023). We found that the deficiency of deglutamylases CCP1 or CCP5 leads to a highly reduced and diffused signal of LCA5, suggesting that the bulge region is lost, thus preventing membrane disc formation and presumably causing photoreceptor death (Figs. 4E,K and 3G,H).

As $Ccp1^{-/-}$ and $Ccp5^{-/-}$ display distal axoneme disorganization, rhodopsin mislocalization throughout the ONL, and lack of LCA5, we hypothesized that these phenotypes could be linked to intraflagellar transport (IFT) defects. Indeed, we recently showed

that the bulge region, marked by LCA5, is crucial to organize IFT at the level of photoreceptor cell, and that the loss of the bulge is associated with defects in the IFT components localization (Faber et al, 2023). In WT, IFT88 accumulates both at the base of the cilium, where trains are formed, and above the CC, at the bulge (Fig. 4F, white arrowheads). By comparison, in both $Ccp1^{-/-}$ and $Ccp5^{-/-}$, the IFT88 signal is greatly reduced above the CC and the cilium entry, highlighting the defects in trafficking towards the photoreceptor outer segment (Fig. 4F,L).

In parallel, we assessed the impact of alpha-tubulin acetyltransferase 1 mutants, which undergo a complete signal loss of acetylation (Appendix Fig. S5). We first show that these mutant mice have no structural defects at the level of the outer segment (Fig. 4A–F), confirming results obtained at the level of the whole retina (Fig. 3I). Moreover, these mutants exhibit no obvious changes in the glutamylation status of the outer segment (Fig. 4A,G). However, we noticed a reduced level of glycylation in $Atat1^{-/-}$ mutants (Fig. 4C,I). The staining of the CC marker—POC5—revealed no difference in the CC length compared to WT (Fig. 4D,J), suggesting that this structure is intact. In line with this, the bulge marker LCA5 was not impacted by the loss of acetylation in the OS (Fig. 4E,K). Finally, in $Atat1^{-/-}$ mutants, we show that the dual localization of IFT88 is conserved (Fig. 4F, white arrowheads) and that the intensity of IFT88 staining is similar to the WT at the base of the cilium (Fig. 4L).

Altogether, we demonstrate that defects in glutamylation, but not acetylation, strongly impact the architecture of the OS axoneme, that is associated with transport defects.

## Progressive disorganization of photoreceptor OS in CCP5 mutant mice

To better define the time course of the CCP5-associated retinal degeneration, we analyzed the anatomy of photoreceptor cilia in 3-, 7-, 10- and 12-month-old $Ccp5^{-/-}$ mice. Using rhodopsin and tubulin staining as readouts for retina integrity, we found that degeneration starts at around 3 months, when ONL is already about

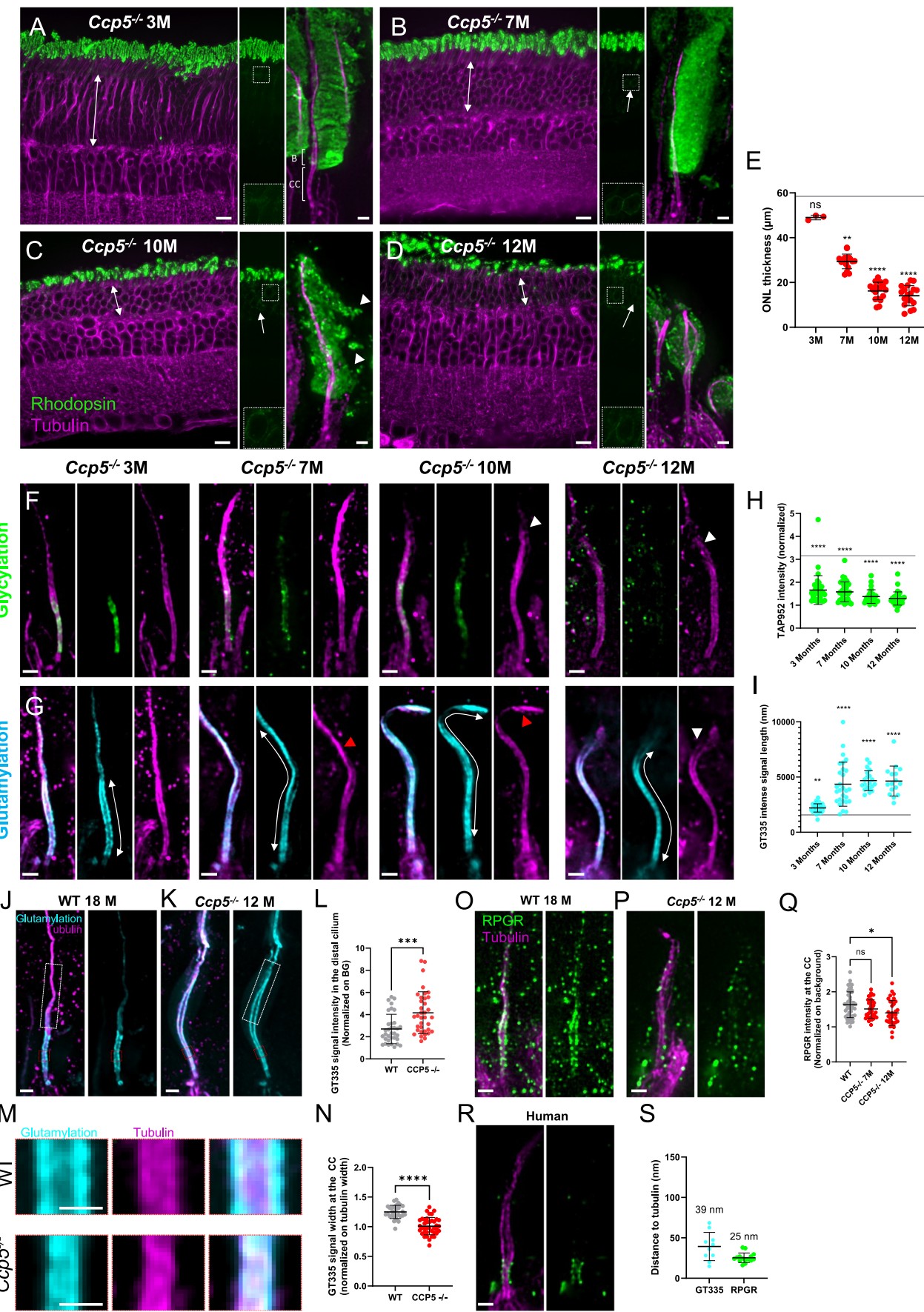

**Figure 5. Dynamic of photoreceptor cell degeneration in CCP5$^{-/-}$ mice.**

(A–D) Overview of photoreceptor cell progressive loss in expanded retina of 3 months- (A), 7 months- (B), 10 months- (C), 12 months- (D) old $Ccp5^{-/-}$ mice stained with rhodopsin (green) and tubulin (magenta). Scale bar: 10 µm. For each time point, a representative image of the rhodopsin channel alone is shown in the middle with an inset and a single photoreceptor cell is depicted on the right. White double arrows reveal the thickness of the ONL measured in (E). White arrows show the rhodopsin signal at the level of the nuclei. White arrowheads show floating rhodopsin signal outside of the outer segment. Scale bar: 500 nm. CC: connecting cilium, B: Bulge. (E) ONL thickness measurement and compared to WT (gray line). $Ccp5^{-/-}$ 3 M: 49.0 nm +/− 1.1 ($n = 3$; $N = 1$); $Ccp5^{-/-}$ 7 M: 29.4 nm +/− 3.3 ($n = 18$; $N = 2$ animals); $Ccp5^{-/-}$ 10 M: 16.2 nm +/− 3.9 ($n = 18$; $N = 2$ animals); $Ccp5^{-/-}$ 12 M: 14.1 nm +/− 4.5 ($n = 18$; $N = 2$ animals) (mean +/− SD). Test: Kruskal–Wallis with Dunn's multiple comparison. WT vs. 3 M: ns (adjusted $P$ value: >0.9999); WT vs. 7 M: **(adjusted $P$ value: <0.0099); WT vs. 10 M: ****(adjusted $P$ value: <0.0001); WT vs. 12 M: ****(adjusted $P$ value: <0.0001). (F, G) Single cell expanded photoreceptor cells of 3 months-, 7 months-, 10 months-, 12 months-old $Ccp5^{-/-}$ mice revealing progressive defects of the outer segment using TAP952 (green) (F), GT335 (cyan) (G) staining. White double arrows depict the length of the signal measured for GT335. Red and white arrowheads point to curled and broken axonemes, respectively. Scale bar: 500 nm. (H) Intensity measurement of TAP952 signal along the outer segment CC normalized on background. $Ccp5^{-/-}$ 3 M: 1.66 +/− 0.63 ($n = 36$; $N = 2$ animals); $Ccp5^{-/-}$ 7 M: 1.58 +/− 0.43 ($n = 34$; $N = 2$ animals); $Ccp5^{-/-}$ 10 M: 1.38 +/− 0.30 ($n = 27$; $N = 1$ animal); $Ccp5^{-/-}$ 12 M: 1.30 +/− 0.29 ($n = 34$; $N = 2$ animals) (mean +/−SD). For each measurement, WT baseline is depicted with a gray line. Measurements were compared to WT values obtained in Fig. 4, and 12-month-old measurements are the same as in Fig. 4. Test: Kruskal–Wallis with Dunn's multiple comparison. WT vs. 3 M: ****(adjusted $P$ value: <0.0001); WT vs. 7 M: ****(adjusted $P$ value: <0.0001); WT vs. 10 M: ****(adjusted $P$ value: <0.0001); WT vs. 12 M: ****(adjusted $P$ value: <0.0001). (I) GT335 intense signal length corrected by the EF. $Ccp5^{-/-}$ 3 M: 2206 nm +/− 374 ($n = 41$; $N = 2$ animals); $Ccp5^{-/-}$ 7 M: 4355 nm +/− 1990 ($n = 28$; $N = 2$ animals); $Ccp5^{-/-}$ 10 M: 4678 nm +/− 903 ($n = 20$; $N = 2$ animals); $Ccp5^{-/-}$ 12 M: 4640 nm +/− 1357 ($n = 16$; $N = 2$ animals) (mean +/− SD). For each measurement, WT baseline is depicted with a gray line. Measurements were compared to WT values obtained in Fig. 4, and 12-month-old measurements are the same as in Fig. 4. Test: Kruskal–Wallis with Dunn's multiple comparison. WT vs. 3 M: **(adjusted $P$ value: 0.0011); WT vs. 7 M: ****(adjusted $P$ value: <0.0001); WT vs. 10 M: ****(adjusted $P$ value: <0.0001); WT vs. 12 M: ****(adjusted $P$ value: <0.0001). (J, K) 18-month-old WT (J) or 12-month-old $Ccp5^{-/-}$ (K) expanded photoreceptor cell stained for glutamylation (GT335, cyan) and tubulin (magenta). White rectangles with dashed lines depict the representative regions used for GT335 intensity measurement in the distal cilium displayed in (L). Red rectangles with dashed lines show the localization of GT335 signal width measurements highlighted in (M) and quantified in (N) Scale bar: 500 nm. (L) Intensity measurement of GT335 in the distal cilium between 18-month-old WT and 12-month-old $Ccp5^{-/-}$ mice normalized on background. WT: 2.70 + /− 1.33 ($n = 34$; $N = 3$); $Ccp5^{-/-}$: 4.16 + /− 1.90 ($n = 36$; $N = 2$) (mean +/− SD). Test: Two-tailed Mann–Whitney test. WT vs. $Ccp5^{-/-}$: ***(adjusted $P$ value: 0.0003). (M) Representative images of GT335 (cyan) signal width at the level of the CC compared to tubulin (magenta). Scale bar: 200 nm. (N) Quantification of GT335 signal width at the CC in 18-month-old WT and 12-month-old $Ccp5^{-/-}$ mice photoreceptor cells normalized on tubulin width. WT: 1.25 +/− 0.11 ($n = 29$; $N = 2$ animals); $Ccp5^{-/-}$: 1.01 +/− 0.15 ($n = 39$; $N = 2$ animals) (mean +/−SD). Test: Two-tailed Mann–Whitney test. WT vs. $Ccp5^{-/-}$: ****(adjusted $P$ value: <0.0001). (O, P) 18-month-old WT (O) or 12-month-old $Ccp5^{-/-}$ (P) expanded photoreceptor cells stained for RPGR (green) and tubulin (magenta). Scale bar: 500 nm. (Q) Quantification of RPGR intensity in adult WT, 7- or 12-month-old $Ccp5^{-/-}$ photoreceptor CC. WT: 1.63 +/− 0.37 ($n = 49$; $N = 3$ animals); $Ccp5^{-/-}$ 7 months: 1.51 +/− 0.26 ($n = 41$; $N = 2$ animals); $Ccp5^{-/-}$ 12 months: 1.40 +/− 0.36 ($n = 34$; $N = 2$ animals) (mean +/− SD). Test: Kruskal–Wallis with Dunn's multiple comparison. WT vs. $Ccp5^{-/-}$ 7 months: ns (adjusted $P$ value: 0.4386); WT vs. $Ccp5^{-/-}$ 12 months: *(adjusted $P$ value: 0.0124). (R) Expanded human photoreceptor cell stained for RPGR (green) and tubulin (magenta). Scale bar: 500 nm. (S) Quantification of GT335 and RPGR signal distances to tubulin in human photoreceptor CC. GT335: 39.25 nm +/− 17.44 ($n = 10$; $N = 1$); RPGR: 25.37 nm +/− 5.97 ($n = 15$; $N = 1$) (mean +/− SD). Each animal corresponds to one experimental replicate. Source data are available online for this figure.

10 µm thinner than in the WT (Fig. 5A,E). ONL thickness progressively decreases in the following months to ultimately reach only 2 or 3 cell layers at 12 months (Fig. 5A–E, white double arrows), highlighting a progressive degeneration over the first year after birth. At 7 months, we noticed a decrease in the ONL thickness to about half of WT, which corresponds to an about 50% decrease in the number of cells observed per field of view with EM (Fig. EV2B). This continuous decrease of the ONL thickness is accompanied by mislocalization of rhodopsin around nuclei, a defect that becomes obvious at 7 months (Fig. 5A–D, white arrows). At the OS level, rhodopsin and tubulin signals reveal the overall absence of defects at 3 months, with membrane discs appearing correctly organized, the bulge region still present, and the distal axoneme straight and properly organized (Fig. 5A). However, from 10 months onwards, we observe a more and more pronounced disorganization of the OS, with opening of axonemes, shorter membrane stacks, and the presence of isolated rhodopsin patches outside the cell (Fig. 5C,D, white arrowheads).

We next analyzed glycylation and glutamylation levels at the OS during the time course of degeneration. Already at 3 months, we find that glycylation signal is strongly reduced concomitantly with highly increased glutamylation (Fig. 5F–I). This tendency further exacerbates over time, with barely any visible glycylation signal at 12 months. At this age, we measured the intensity of GT335 signal in the distal cilium and confirmed that the glutamylation propagates towards the distal part of the cilium in $Ccp5^{-/-}$ mice (Fig. 5J–L). Interestingly, the external GT335 signal that forms a sheath-like structure at the level of the CC in the WT is lost in $Ccp5^{-/-}$ mice, with most GT335 signal

colocalizing with tubulin (Fig. 5J,K,M,N). This result suggests that rather than tubulin, another glutamylated protein generating this external signal at the CC is lost or relocalized in $Ccp5^{-/-}$ mice, possibly participating in the collapse of the photoreceptor cell OS. In line with that, we tested RPGR, known to be glutamylated by TTLL5 and recognized by the GT335 antibody (Sun et al, 2016). We stained for this protein in WT and $Ccp5^{-/-}$ mice (Fig. 5O,P). In the WT, RPGR seems to localize in both the CC and the bulge region. Interestingly, we observed a progressive decrease in RPGR signal in $Ccp5^{-/-}$ mice, suggesting that the loss of the sheath-like GT335 signal could be associated with the loss of RPGR (Fig. 5Q). This protein being associated with IFT transport at the level of the CC (Hosch et al, 2011), our observation is consistent with the IFT defects observed in $Ccp5^{-/-}$ photoreceptor cells. Finally, to confirm RPGR localization, we also stained RPGR in human photoreceptors, and observed that the signal clearly localizes at the level of the CC and the proximal part of the bulge, and external to the tubulin, where the GT335 sheath-like is observed (Fig. 5R,S), highlighting the conservation of this pattern.

To obtain complementary insights into the molecular mechanisms involved in CCP5-associated retinitis pigmentosa, we then analyzed CC and IFT markers during the time course of degeneration (Fig. 6A–F). At 3 months, both the CC length (POC5) (Fig. 6A,C) and the IFT enrichment at the basal body (IFT88) (Fig. 6B,D) are comparable to the WT, suggesting that the PTM imbalance (Fig. 5F–I) precedes the structural and functional defects of the OS. From 7 months onwards and gradually increasing up to 12 months, we find that axoneme disorganization is more and more pronounced, with distal axonemal microtubules mostly

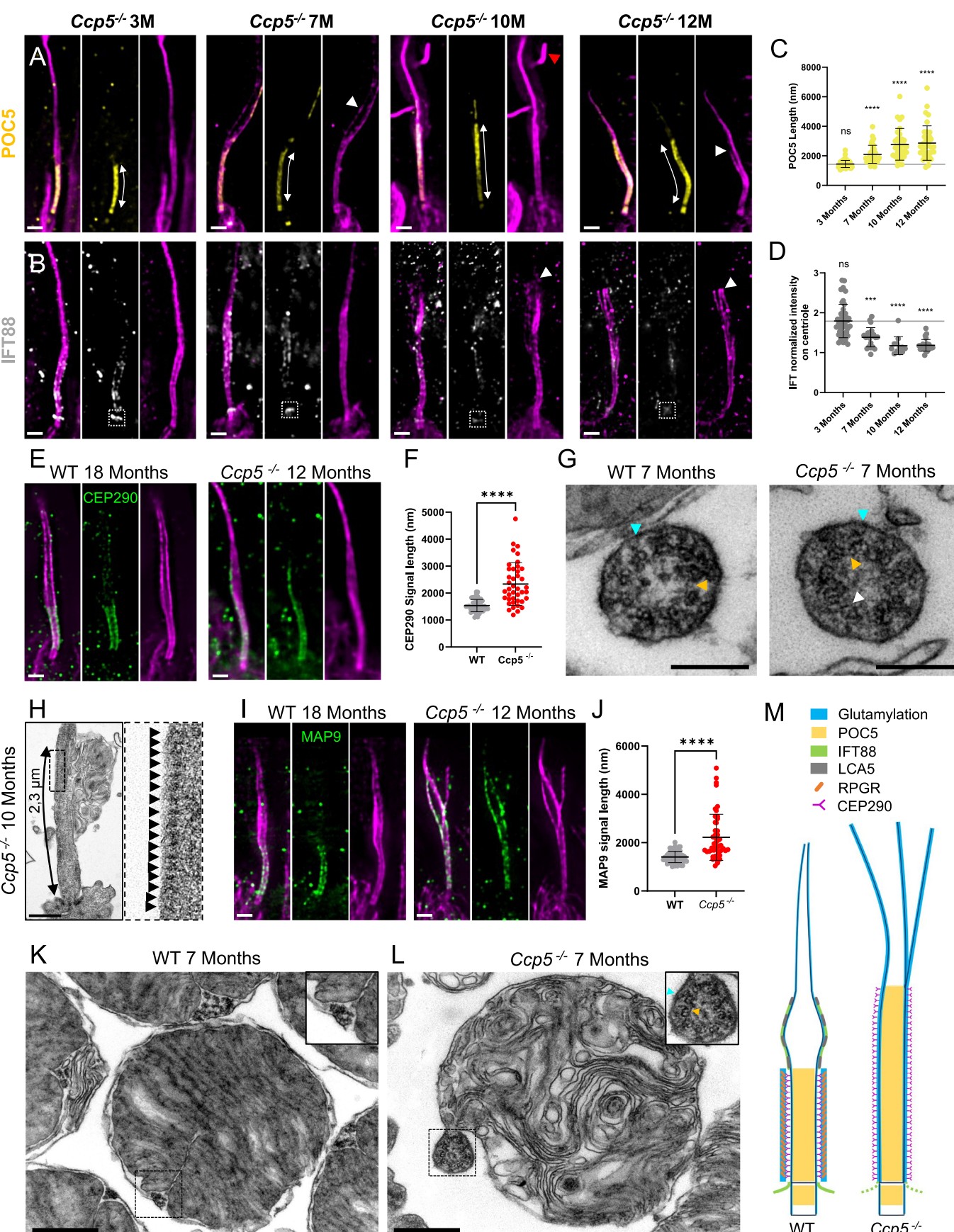

◀

**Figure 6. Connecting cilium exacerbation and IFT transport defects in CCP5 $^{-/-}$ mice.**

(A, B) Single cell expanded photoreceptor cells of 3 months, 7 months-, 10 months-, 12 months-old *Ccp5*$^{-/-}$ mice highlighting CC (POC5, yellow) (A), and IFT (IFT88, white) defects. White double arrows depict the length of the signal measured for POC5. Red and white arrowheads point to curled and broken axonemes, respectively. Rectangles with dashed lines show the area where intensity measurements were picked with IFT88 signal. Scale bar: 500 nm. (C) CC length measured with POC5 and corrected by the EF. *Ccp5*$^{-/-}$ 3 M: 1446.0 nm +/− 241 ($n = 61$; $N = 2$ animals); *Ccp5*$^{-/-}$ 7 M: 2100 nm +/− 609 ($n = 38$; $N = 1$ animal); *Ccp5*$^{-/-}$ 10 M: 2779 nm +/− 1076 ($n = 31$; $N = 1$ animal); *Ccp5*$^{-/-}$ 12 M: 2864 nm +/− 1164 ($n = 36$; $N = 2$ animals) (mean +/−SD). Test: Kruskal–Wallis with Dunn's multiple comparison. WT vs. 3 M: ns (adjusted *P* value: >0.9999); WT vs. 7 M: ****(adjusted *P* value: <0.0001); WT vs. 10 M: ****(adjusted *P* value: <0.0001); WT vs. 12 M: ****(adjusted *P* value: <0.0001). For each measurement, WT baseline is depicted with a gray line. Measurements were compared to WT values obtained in Fig. 4, and 12 month-old measurements are the same as in Fig. 4. (D) IFT88 intensity at the ciliary base normalized on background. *Ccp5*$^{-/-}$ 3 M: 1.80 + /− 0.42 ($n = 44$; $N = 2$ animals); *Ccp5*$^{-/-}$ 7 M: 1.36 + /− 0.24 ($n = 22$; $N = 2$ animals); *Ccp5*$^{-/-}$ 10 M: 1.17 + /− 0.23 ($n = 12$; $N = 1$ animal); *Ccp5*$^{-/-}$ 12 M: 1.18 + /− 0.15 ($n = 25$; $N = 2$ animals) (mean +/−SD). Test: Kruskal–Wallis with Dunn's multiple comparison. WT vs. 3 M: ns (adjusted *P* value: >0.9999); WT vs. 7 M: ***(adjusted *P* value: 0.0004); WT vs. 10 M: ****(adjusted *P* value: <0.0001); WT vs. 12 M: ****(adjusted *P* value: <0.0001). For each measurement, WT baseline is depicted with a gray line. Measurements were compared to WT values obtained in Fig. 4, and 12 month-old measurements are the same as in Fig. 4. (E) 18-month-old WT or 12-month-old *Ccp5*$^{-/-}$ expanded photoreceptor cell stained for CEP290 (green) and tubulin (magenta). Scale bar: 500 nm. (F) CEP290 signal length in 18-month-old WT or 12-month-old *Ccp5*$^{-/-}$ photoreceptor cell OS corrected by the EF. WT: 1534 nm +/− 230 ($n = 46$; $N = 2$ animals); *Ccp5*$^{-/-}$: 2334 nm +/− 792 ($n = 41$; $N = 2$ animals) (mean +/− SD). Test: Two-tailed Mann–Whitney test. WT vs. *Ccp5*$^{-/-}$: ****(adjusted *P* value: <0.0001). (G) EM micrographs representing transverse sections of the CC in 7-month-old WT or *Ccp5*$^{-/-}$ photoreceptor cells. Blue and orange arrowheads show the presence of Y-links, the inner scaffold, respectively. The two micrographs are also represented in the gallery in Fig. EV4. Scale bar: 200 nm. (H) On the left, EM image showing a longitudinal view of a 10-month-old *Ccp5*$^{-/-}$ photoreceptor outer segment. Black double arrow shows the length between the beginning of the CC and the last Y-link structure observed. An inset of the rectangle with dashed lines is represented on the right to highlight the presence and the periodicity of the Y-links. Average distance between consecutive Y-links is 36,9 nm. Scale bar: 500 nm. (I) 18-month-old WT or 12-month-old *Ccp5*$^{-/-}$ expanded photoreceptor cell stained for MAP9 (green) and tubulin (magenta). Scale bar: 500 nm. (J) MAP9 signal length in 18-month-old WT or 12-month-old *Ccp5*$^{-/-}$ photoreceptor cell OS corrected by the EF. WT: 1407 nm +/− 235 ($n = 68$; $N = 3$ animals); *Ccp5*$^{-/-}$: 2221 nm +/− 956 ($n = 49$; $N = 2$ animals) (mean +/− SD). Test: Two-tailed Mann–Whitney test. WT vs. *Ccp5*$^{-/-}$: ****(adjusted *P* value: <0.0001). (K, L) EM micrograph of 7-month-old WT (K) or *Ccp5*$^{-/-}$ (L) outer segment. An inset highlighting the difference of the axonemes is depicted on the upper right of the image. Blue or orange arrowheads show the presence of Y-links or the inner scaffold, respectively. Note the unstructured organization of the membrane discs in the mutant compared to the WT. Scale bar: 500 nm. (M) Model summarizing the defects observed in a *Ccp5*$^{-/-}$ photoreceptor cell outer segment compared to WT. Loss of CCP5 leads to hyperglutamylation (cyan) that propagates towards the distal part of the cilium. This is accompanied by the loss of RPGR (orange), the bulge region delineated by LCA5 (gray), the elongation of the CC marked with POC5 (yellow) and Y-links (magenta), and IFT defects (green). Consequently, the distal part of the axoneme is disorganized. Each animal corresponds to one experimental replicate. Source data are available online for this figure.

curled or broken (Figs. 6A,B and 5F,G red and white arrowheads, respectively) whereas CC are mostly preserved. IFT88 intensity is progressively reduced and ultimately absent from 12-month-old photoreceptor cells (Fig. 6B,D). In parallel, the length of the CC gradually increases (Fig. 6A,C). We confirmed this result in 12-month-old *Ccp5*$^{-/-}$ mice using another CC marker, CEP290, which has been proposed to form the Y-links bridging microtubule doublets (MTDs) to the membrane periodically along the CC (Zhang et al, 2024; Louvel et al, 2023) (Fig. 6E,F). To assess the integrity of the CC axoneme at the ultrastructure level, we examined electron microscopy images of 7-month-old *Ccp5*$^{-/-}$ photoreceptor cells, focusing on transverse sections of CC (Figs. 6G and EV4). In both WT and *Ccp5*$^{-/-}$ CC, it was possible to identify the presence of Y-links (Fig. 6G, blue arrowheads). We also confirmed the presence of the inner scaffold (Mercey et al, 2022) (Fig. 6G, orange arrowheads), confirming that CC in *Ccp5*$^{-/-}$ photoreceptor cells keep their structural integrity. However, in a few cases, we observed that the B-tubules appear open in mutant CC, a feature never seen in the WT, but this observation might also arise from the low contrast observed in the mutants (Fig. EV4, white arrowheads). Interestingly, electron microscopy images of 10-month-old *Ccp5*$^{-/-}$ photoreceptor cells revealed the presence of Y-links, with a periodicity of 37 nm as previously observed (Zhang et al, 2024), on a distal portion of the cilium where the CC should be terminated, corroborating the extended CEP290 signal observed in expansion microscopy in the same mutants (Fig. 6H, black arrowheads).

We finally examined the localization of MAP9, which has recently been described to be associated with MTDs and partly dependent on glutamylation levels (Tran et al, 2024) (Fig. 6I,J). While in the WT, MAP9 signal decorates the CC, the signal is propagated toward the distal part of the cilium in *Ccp5*$^{-/-}$ mice,

similarly to what we observed for CEP290 and POC5. These results suggest that axonemal ultrastructure in the outer segment could be impaired in *Ccp5*$^{-/-}$ mutant mice, with the presence of MTD-associated structures (inner scaffold and Y-links). To confirm this hypothesis, we compared 7-month-old WT and *Ccp5*$^{-/-}$ mice using EM (Fig. 6K,L). In the WT OS, microtubules are packed into the membrane incisure cavity, with no round shape organization compared to the CC due to the absence of inner scaffold and Y-links (Fig. 6K, inset). By contrast, in *Ccp5*$^{-/-}$ OS, we observed organized MTDs, with the presence of both inner scaffold and Y-links (Fig. 6L, inset), corroborating the extension of the CC observed with POC5 and CEP290 staining (Fig. 6A,E,H). We also noticed that in the mutant, the axoneme is mostly disconnected from the membrane discs, themselves highly impaired.

Altogether, we showed that *Ccp5*$^{-/-}$ mice exhibit a slow and progressive degeneration of the photoreceptor cells characterized by the progressive decrease of glycylation and a concomitant increase of hyperglutamylation. At later stages, the signal of ciliary transport proteins is progressively reduced, in parallel with a loss of RPGR and a disorganization of distal axonemal microtubules. The loss of the bulge region accompanied by the exacerbation of the CC is presumably leading to the inability to form new membrane discs, ultimately causing photoreceptor cell death (Fig. 6M).

## Discussion

The photoreceptor outer segment, reaching a length of 50 μm in the human eye, is organized around its microtubule-based axoneme, and provides structural support for the regularly stacked membrane discs. As we recently described, disorganization of the axonemal structure leads to massive outer segment collapse, causing

photoreceptor death (Mercey et al, 2022). Therefore, cellular determinants assuring the integrity of axonemal microtubules are expected to be of prime importance for the function and survival of photoreceptor cells. Besides structural features such as the inner scaffold inside the connecting cilium that directly maintains axoneme cohesion by connecting neighboring microtubule doublets (Mercey et al, 2022), tubulin PTMs have emerged as molecular actors of microtubule stability and function. The importance of tubulin PTMs in photoreceptor maintenance has recently been highlighted by the fact that mutations of *AGBL5*, coding the deglutamylase CCP5, lead to retinitis pigmentosa in human. However, mechanisms of photoreceptor degeneration associated with CCP5 deficiency have remained unknown.

Here, we explored the molecular localization of PTMs and assessed consequences of PTMs perturbations for the photoreceptor outer segment. We first provided a molecular mapping of 4 different tubulin PTMs: glycylation, acetylation, glutamylation (mono- and poly-) and detyrosination and revealed that they form distinct patterns along the outer segment. At the level of the connecting cilium, all the analyzed PTMs are present, with different localizations. Whereas acetylation and glycylation are observed on the microtubules, as expected, we were surprised to see that glutamylation and detyrosination exhibit a strong signal, restricted to the CC, but about 60 nm away from the microtubule signal center of mass.

This distance and the absence of tubulin staining at this location excluded the possibility that we detect microtubule or tubulin trafficking along the CC. Therefore, glutamylation and detyrosination might decorate other substrates, particularly enriched along the inner part of the CC membrane. It has been previously shown that the protein RPGR, a CC component, is glutamylated and is recognized by the GT335 antibody (Sun et al, 2016). Our demonstration that RPGR signal in human photoreceptors is overlapping with the GT335-positive sheath-like signal at the CC strongly suggests that this external GT335 signal indeed corresponds to glutamylated RPGR. However, the antibody against detyrosinated tubulin is supposed to recognize specifically the C-terminal sequence of alpha-tubulin. Why only the CC reveals such a pattern of detyrosination remains unknown, but we cannot exclude nonspecific signal.

We next analyzed the effect of PTM perturbation on photoreceptor cell maintenance, focusing on glutamylation, since mutations of the deglutamylase CCP5 lead to retinitis pigmentosa in human. Loss of either CCP1 or CCP5 deglutamylases leads to retinal degeneration in few months, where only 2 or 3 layers of photoreceptor nuclei remain at about one year of age (vs about 10 layers in WT, Fig. EV2A). We used U-ExM to describe the degeneration process at the cellular level. Interestingly, the first obvious phenotype observed in these two mutant mice is the disorganization of the axoneme, that occurs above the CC (Fig. 3G,H). This is distinct from what we described for *Fam161a* mutation (also leading to retinitis pigmentosa), where microtubules spread just above the basal body (Mercey et al, 2022), thus indicating a different molecular mechanism. The fact that the CC is mostly preserved from microtubule collapse in *Ccp1*$^{-/-}$ and *Ccp5*$^{-/-}$ mice is highly similar to what we previously observed for the deficiency of the bulge protein Lebercilin (LCA5), causing Leber Congenital Amaurosis in human (Faber et al, 2023). Intriguingly, in Lebercilin-deficient mice, the bulge is no longer present, and CC

markers exhibit longer signals, suggesting that Lebercilin could act as a ruler to dictate CC length. In *Ccp5*$^{-/-}$ mice, CC size is even more exacerbated, and LCA5 is no longer present, suggesting similar mechanisms in these two mutants, even if the onset of the degeneration is faster in LCA5 deficient mice (within the first month after birth).

We further showed that CCP5 loss leads to an important hyperglutamylation that is paralleled with the loss of glycylation, highlighting the competition between these two PTMs, as previously described for mice lacking glycylation in the retina (Grau et al, 2017) (Fig. 5H). Importantly, hyperglutamylation is not restricted to the outer segment in *Ccp5*$^{-/-}$, as the whole inner segment exhibits a strongly increased GT335 signal. We cannot exclude that hyperglutamylation of cytosolic microtubules in the cell body is partly responsible for photoreceptor cell death. Interestingly, defects of TTLL5, a glutamylase, leading to hypoglutamylation on its substrates is also leading to photoreceptor degeneration (Sun et al, 2016), showing that the correct adjustment of physiological glutamylation levels is crucial to maintain the correct function of photoreceptor cells.

Surprisingly, direct comparison of *Ccp1*$^{-/-}$ and *Ccp5*$^{-/-}$ mice at late stage revealed distinct effects on CC and on hyperglutamylation level. One reason for this might be the time course of degeneration between *Ccp1*$^{-/-}$ and *Ccp5*$^{-/-}$ mice. CCP1 degeneration seems faster compared to CCP5, as the ONL thickness is comparable between 8-month-old *Ccp1*$^{-/-}$ mice and 12-month-old *Ccp5*$^{-/-}$ mice (Fig. 3E). Moreover, *Ccp1*$^{-/-}$ photoreceptor outer segments are shorter at 8-months compared to *Ccp5*$^{-/-}$, reflecting a more advanced degeneration, where only a small portion of the axoneme is remaining. This would explain the difference in POC5 and GT335 signal length between *Ccp1*$^{-/-}$ and *Ccp5*$^{-/-}$ outer segments.

It has been shown for various types of cilia that impaired glutamylation leads to ultrastructural defects of the B-tubule, possibly impairing intraflagellar transport (IFT) (Yang et al, 2021). In line with that, we revealed by EM occasional cases of open B-tubules in *Ccp5*$^{-/-}$ photoreceptor CC (Fig. EV4). Furthermore, we observed a loss of IFT88 signal in both *Ccp1*$^{-/-}$ and *Ccp5*$^{-/-}$ mice at the base of the cilium, similarly to what we previously showed in *Lca5* mutant mice (Faber et al, 2023) (Fig. 4F). This result suggests that hyperglutamylation could impair IFT transport towards the bulge region, where membrane discs form, leading to the progressive collapse of the outer segment. We previously showed that in WT, IFT components are enriched at the bulge region, concentrating building blocks to form membrane discs mice (Faber et al, 2023). Since LCA5 signal at the bulge is also lost in deglutamylase mutants, a possible explanation is that hyperglutamylation leads to the loss of the bulge region, thus causing IFT components to diffuse to the distal cilium and preventing their recycling. Interestingly, we showed that in LCA5 mutant mice, glutamylation signal is also seen along the distal axoneme, suggesting that in physiological conditions, LCA5 could prevent hyperglutamylation at the level of the bulge and above (Fig. EV5). We also demonstrated that in *Ccp5*$^{-/-}$ photoreceptor cells, RPGR is lost at the level of the CC. It has been recently shown that RPGR regulates actin dynamics at the bulge region, crucial for membrane disc formation (Megaw et al, 2024). That could explain, at least in part, why membrane discs are disorganized in mutant mice, participating to the OS collapse. Altogether, these results suggest

that hyperglutamylation-associated OS collapse could result from the combination of (i) the loss of the bulge region, (ii) the extension of the connecting cilium region with elongated inner scaffold and Y-links coverage, (iii) the loss of RPGR and (iv) defective IFT distribution. However, whether these defects are all directly related to a perturbed glutamylation balance remains to be elucidated.

Our demonstration that tubulin PTMs are similarly present and distributed in human retina strongly suggests that observations made in the mouse models are relevant for human. One important difference we observed is the length of the connecting cilia, which in human is half the length of the mouse. This seems counter-intuitive given that the outer segment of photoreceptors is longer in human as compared to mouse. A careful analysis in several species would help to understand how the length of the CC is regulated. A recent study analyzing different markers of canine photoreceptor OS using U-ExM showed that CC length in rods is slightly smaller compared to the mouse (Takahashi et al, 2024).

Altogether, our study revealed the importance of controlled levels of glutamylation in the highly specialized primary cilium of photoreceptor cells, the outer segment, to maintain the integrity of the axonemal structure. In addition, this work highlights the need to elucidate the subcellular events involved in retinal diseases such as retinitis pigmentosa. Indeed, this pathology being associated with mutations in about 80 genes, it is crucial to understand specific molecular mechanisms linked to each gene, to properly adapt therapeutic options to cure or slow down this type of diseases.

# Methods

### Reagents and tools table

| Reagent/resource | Reference or source | Identifier or catalog number |
|---|---|---|
| **Experimental models** | | |
| (*M. musculus*) C57BL/6 | Ccp1-/- (PMID: 29449678), Ccp5-/- (PMID: 30635446), Atat1-/- (PMID: 23748901), Lca5-/- (PMID: 37071472). | |
| Healthy human tissue not used for diagnostic procedure | Hôpital Ophtalmologique Jules Gonin, Lausanne, Switzerland | |
| **Antibodies** | | |
| Rabbit anti CEP290 | Proteintech | 22490-1-AP |
| Mouse anti Rhodopsin | Thermo Fisher | MA5-11741 |
| Mouse anti B-tubulin | ABCD antibodies | AA344 - scFv-S11B |
| Mouse anti A-tubulin | ABCD antibodies | AA345 - scFv-F2C |
| Rabbit anti POC5 | Bethyl | A303-341A |
| Rabbit anti LCA5 | Proteintech | 19333-1-AP |
| Mouse anti TAP952 | Merck Millipore | MABS277 |
| Mouse anti Acetylated tubulin | Thermo Fisher | 32-2700 |
| Mouse anti GT335 | Adipogen | AG-20B-0020 |
| Rabbit anti PolyE | Adipogen | AG-25B-0030 |

| Reagent/resource | Reference or source | Identifier or catalog number |
|---|---|---|
| Rabbit anti detyrosinated tubulin | RevMab biosciences | 31-1335-00 |
| Rabbit anti IFT88 | Proteintech | 13967-1-AP |
| Rabbit anti Delta2 tubulin | Merck Millipore | AB3203 |
| Rabbit anti RPGR | Proteintech | 16891-1-AP |
| Rabbit anti RPGR | Merck–Sigma Aldrich | HPA001593 |
| Rabbit anti MAP9 | Proteintech | 26078-1-AP |
| **Chemicals, enzymes and other reagents** | | |
| Bis-acrylamide (BIS) | Merck–Sigma Aldrich | M1533 |
| Acrylamide 40% w/w | Merck–Sigma Aldrich | A4058 |
| Formaldehyde 35-38% | Merck–Sigma Aldrich | F8775 |
| Sodium acrylate | AK Scientific | R624 |
| Ammonium persulfate (APS) | Thermo Fisher | 17874 |
| Tetramethylethylenediamine (TEMED) | Thermo Fisher | 17919 |
| Poly-D-Lysine | Merck–Sigma Aldrich | A38904-01 |
| Sodium dodecyl sulfate | Pan Reac Applichem | A7219 |
| Tris-Base | Roth | 2449.3 |
| Tween 20 | Roth | 9127-2 |
| Bovine Serum Albumin (BSA) | Merck–Sigma Aldrich | 10735086001 |
| Paraformaldehyde 16% | Electron Microscopy Science | 15710 |
| Glutaraldehyde 25% | Electron Microscopy Science | 16200 |
| 2% Osmium tetraoxide | Merck–Sigma Aldrich | 05500-1g |
| Uranyl acetate | Polysciences | 21447-25g |
| propylene oxide | Merck–Sigma Aldrich | 82320-1L |
| **Software** | | |
| Fiji | https://imagej.net/software/fiji/downloads | |
| GraphPad Prism 10 | https://www.graphpad.com | |
| **Other** | | |
| 35 mm Petri dish | MatTek | P35G-1.5-10-C |
| **Microscopes** | | |
| G2 Sphera | | |
| Leica Thunder DMi8 | | |
| Leica Stellaris 8 | | |

## Methods and protocols

### Mutant mouse models

Animal care and use for this study were performed in accordance with the recommendations of the European Community (2010/63/UE) for the care and use of laboratory animals. Experimental procedures were specifically approved by the Ethics Committee of the Institut Curie CEEA-IC #118 (APAFIS #37315-2022051117455434 v2) in compliance with the international guidelines.

Mice used in this study have been described before: $Ccp1^{-/-}$ (PMID: 29449678), $Ccp5^{-/-}$ (pmid: 30635446), $Atat1^{-/-}$ (PMID: 23748901).

### Human tissue

The use of human samples was approved by the local Ethics Committee (CER-VD protocol No. 340-15) and the patients signed informed consent.

### Ultrastructure expansion microscopy (U-ExM) of mouse and human retinas

Mice of desired age and genotype were sacrificed by cervical dislocation and their eyes were immediately collected and immersed in 4% PFA in PBS. They were incubated overnight at 4 °C, washed 3 ×20 min in PBS at room temperature and then stored at 4 °C in PBS until required.

Human retina tissue was taken from a healthy retinal region of enucleated eyes due to tumor exenteration. Retina sample was fixed for 60 min in 4% PFA at RT and then wash in PBS before preparation for U-ExM.

Retina were then processed as described elsewhere (Mercey et al, 2022; Faber et al, 2023). Briefly, once flattened, retinas were placed inside the well of a 35 mm Petri dish for U-ExM processing. Retinas were first incubated overnight (ON) in 100 μL of 2% acrylamide + 1.4% formaldehyde at 37 °C. The next day, solution is removed and 35 μL monomer solution composed of 25 μL of sodium acrylate (stock solution at 38% [w/w] diluted with nuclease-free water), 12.5 μL of AA, 2.5 μL of N,N′-methylenebisacrylamide (BIS, 2%), and 5 μL of 10× PBS was added for 90 min at RT. Then, MS was removed and 90 μL of MS was added together with ammonium persulfate (APS) and tetra-methylethylenediamine (TEMED) as a final concentration of 0.5% for 45 min at 4 °C first followed by 3 h incubation at 37 °C to allow gelation. A 24-mm coverslip was added on top to close the chamber. Next, the coverslip was removed and 1 ml of denaturation buffer (200 mM SDS, 200 mM NaCl, 50 mM Tris Base in water (pH 9)) was added into the MatTek dish for 15 min at RT with shaking. Then, careful detachment of the gel from the dish with a spatula was performed, and the gel was incubated in 1.5 ml tube filled with denaturation buffer for 1 h at 95 °C and then ON at RT. The day after, the gel was cut around the retina that is still visible at this step and expanded in three successive ddH2O baths. Then, the gel was manually sliced with a razorblade to obtain ~0.5 mm thick transversal sections of the retina that were then processed for immunostaining.

### Immunostainings

Gel slices were first incubated in three successive PBS 1× baths of 5 min. Then, gels were incubated with primary antibodies (Reagent and Tools Table) in PBS with 2% of bovine serum albumin (BSA) overnight at 4 °C. Gels were then washed three times 5 min in PBS with 0.1% Tween 20 (PBST) prior to secondary antibodies incubation for 3 h at 37 °C. After a second round of washing (3 times 5 min in PBST), gels were expanded with three 10-min baths of ddH20 before imaging. Image acquisition was performed on an inverted confocal Leica Stellaris 8 microscope or on a Leica Thunder DMi8 microscope using a 20× (0.40 NA) or 63× (1.4 NA) oil objective with Lightning or Thunder SVCC (small volume computational clearing) mode at max resolution, adaptive as "Strategy" and water as "Mounting medium" to generate deconvolved images. 3D stacks were acquired with 0.12 μm z-intervals and an x, y pixel size of 35 nm.

### Electron microscopy

Retina cups were first incubated overnight at RT with 3% PFA and 0.1% glutaraldehyde in PBS. Samples were further treated with 2% osmium tetroxide in buffer for 30 min and immersed in a solution of uranyl acetate 0.25% overnight to enhance contrast of membranes. Samples were dehydrated in increasing concentrations of ethanol followed by pure propylene oxide, and then embedded in Epon resin. Serial ultrathin sections of 50 nm were finally cut and stained with 5% uranyl acetate (in H$_2$O) and Reynolds' lead citrate. Micrographs were acquired using a G2 Sphera microscope operated at 120 kV equipped with an Eagle detector at two magnifications: low magnification of ×3200 corresponding to a pixel size of 22.8 nm and high magnification of ×42,000 corresponding to a pixel size of 1.91 nm. Location of the different sections along the proximal to distal OS axis was determined thanks to the shape of the plasma membrane and the presence of different structures (Y-Links, Inner scaffold).

### Quantifications

**Expansion factor**: The expansion factor was calculated in a semiautomated way by comparing the full width at half maximum (FWHM) of photoreceptor mother centriole proximal tubulin signal with the proximal tubulin signal of expanded human U2OS cell centrioles using PickCentrioleDim plugin described elsewhere (Borgne et al, 2022). Briefly, more than 50 photoreceptor mother centrioles FWHM were measured and compared to a pre-assessed value of U2OS centriole width (25 centrioles: mean = 231.3 nm +/− 15.6 nm). The ratio between measured FWHM and known centriole width gave the expansion factor (Fig. EV1C).

**ONL thickness**: ONL thickness was measured manually using tubulin staining on at least two different ×20 original magnification images per replicate. Three measurements were performed per image to avoid bias due to retina dissection or slicing. Each measurement was subsequently corrected for the expansion factor.

**Protein diameter**: Using ImageJ, a line crossing centriole or connecting cilia on their diameter was drawn and plot profiles of each channel (protein of interest and tubulin) were generated. Then, distances between peak intensities of each protein were recorded. Average tubulin diameter was set at 170 nm from cryo ET data (Robichaux et al, 2019), to generate an expansion factor value for each protein measurement.

**Protein signal width**: GT335 and tubulin signal widths were calculated in a semiautomated way using PickCentrioleDim plugin described elsewhere (Borgne et al, 2022). Photoreceptor CC FWHM were measured for tubulin and GT335 signals and the ratio was calculated by dividing GT335 FWHM by tubulin FWHM.

**Protein length**: Protein signal lengths were measured using a segmented line drawn by hand (FIJI) to fit with photoreceptor curvature and corrected with the expansion factor.

**Intensity measurements**: Fluorescence intensity measurements of TAP952, LCA5, GT335 and IFT88 were performed on maximal projections using FIJI on denoised images. The same rectangular region of interest (ROI) drawn by hand was used to measure the mean gray value of the protein signal and the corresponding background. Fluorescence intensity was finally calculated by dividing the mean gray value of the fluorescence signal by the mean gray value of the background (normalized mean gray value). For TAP952, measurements were performed all along photoreceptor CC and bulge, defined by tubulin. For LCA5 and IFT88, measurements were performed on the bulge region, and the basal body, respectively, defined by tubulin.

**Statistics**: The comparisons of more than two groups were made using nonparametric Kruskal–Wallis test followed by post hoc test (Dunn's for

multiple comparisons) to identify all the significant group differences. Every measurement was performed on at least two different animals, unless specified. Data are all represented as a scatter dot plot with centerline as mean, except for percentages quantifications, which are represented as histogram bars. The graphs with error bars indicate SD ($+/-$) and the significance level is denoted as usual (*$P < 0.05$, **$P < 0.01$, ***$P < 0.001$, ****$P < 0.0001$). All the statistical analyses were performed using Prism9. When possible, a minimum of 10 measurements have been performed per animal.

Note that $P$ values below 0.0001 or above 0.9999 are annotated as $P < 0.0001$ or $P > 0.9999$.

## Data availability

No data was used for the research described in the article.

The source data of this paper are collected in the following database record: biostudies:S-SCDT-10_1038-S44318-024-00284-1.

## Peer review information

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

## Acknowledgements

This work was funded by the Swiss National Science Foundation (SNSF) grants 310030_205087 (VH, PG), the ProVisu and Gelbert foundations (VH, CK). CJ is supported by the French National Research Agency (ANR) awards ANR-20-CE13-0011, ANR-21-CE14-0045, and the Fondation pour la Recherche Medicale (FRM) grant MND202003011485. We thank the BioImaging Center of the University of Geneva, as well as K Belloul, V Dangles-Marie, V Henriot, C Jouhanneau (Institut Curie) for technical assistance. We thank Ronald Roepman for the use of $Lca5^{gt/gt}$ mouse. We also thank the electron microscopy facility (PFMU) in Geneva.

## Author contributions

**Olivier Mercey**: Formal analysis; Validation; Investigation; Visualization; Methodology; Writing—original draft; Writing—review and editing. **Sudarshan Gadadhar**: Methodology. **Maria M Magiera**: Resources; Methodology; Writing—review and editing. **Laura Lebrun**: Methodology. **Corinne Kostic**: Resources; Writing—review and editing. **Alexandre Moulin**: Resources. **Yvan Arsenijevic**: Resources; Writing—review and editing. **Carsten Janke**: Supervision; Funding acquisition; Writing—review and editing. **Paul Guichard**: Conceptualization; Supervision; Funding acquisition; Writing—original draft; Project administration; Writing—review and editing. **Virginie Hamel**:

Conceptualization; Supervision; Funding acquisition; Writing—original draft; Project administration; Writing—review and editing.

Source data underlying figure panels in this paper may have individual authorship assigned. Where available, figure panel/source data authorship is listed in the following database record: biostudies:S-SCDT-10_1038-S44318-024-00284-1.

## Disclosure and competing interests statement

The authors declare no competing interests.

# Expanded View Figures

**Figure EV1.  Architecture of the photoreceptor outer segment.**

(**A**) Expanded photoreceptor cell layer of a WT mouse retina stained for tubulin (magenta) and rhodopsin (green). On the right, inset of a single photoreceptor cell outer segment stained for tubulin, revealing the different regions of the cilium. Scale bars: left: 10 μm; right: 500 nm. (**B**) Quantification of the expansion factor (EF) used for the whole study. EF = 4.248 + / − 0.588 (mean +/− SD) ($n = 55$; $N > 5$ animals). (**C**) Model explaining the tubulin code and highlighting the PTMs analyzed in this study. (**D**) 18-month-old WT expanded photoreceptor cell stained for glutamylation (GT335, cyan) and tubulin (magenta) highlighting differences in GT335 staining along the OS. Scale bar: 500 nm. (**E**) Zoom in on the 3 different OS subregions of GT335 staining analyzed (Centriole, CC, Bulge). Scale bar: 200 nm. (**F**) Quantification of GT335 signal intensity in the centriole, the CC and the bulge region, normalized on background. Centriole: 5.55 + / − 3.61 ($n = 38$); CC: 15.12 + / − 12.67 ($n = 39$); Bulge: 8.08 + / − 6.00 ($n = 39$) ($N = 3$ animals) (mean +/− SD). Test: Kruskal–Wallis with Dunn's multiple comparison. centriole vs. CC: ****(adjusted $P$ value: <0.0001); centriole vs. bulge: ns (adjusted $P$ value: 0.3331); CC vs. bulge: *(adjusted $P$ value: 0.0272). Each animal corresponds to one experimental replicate.

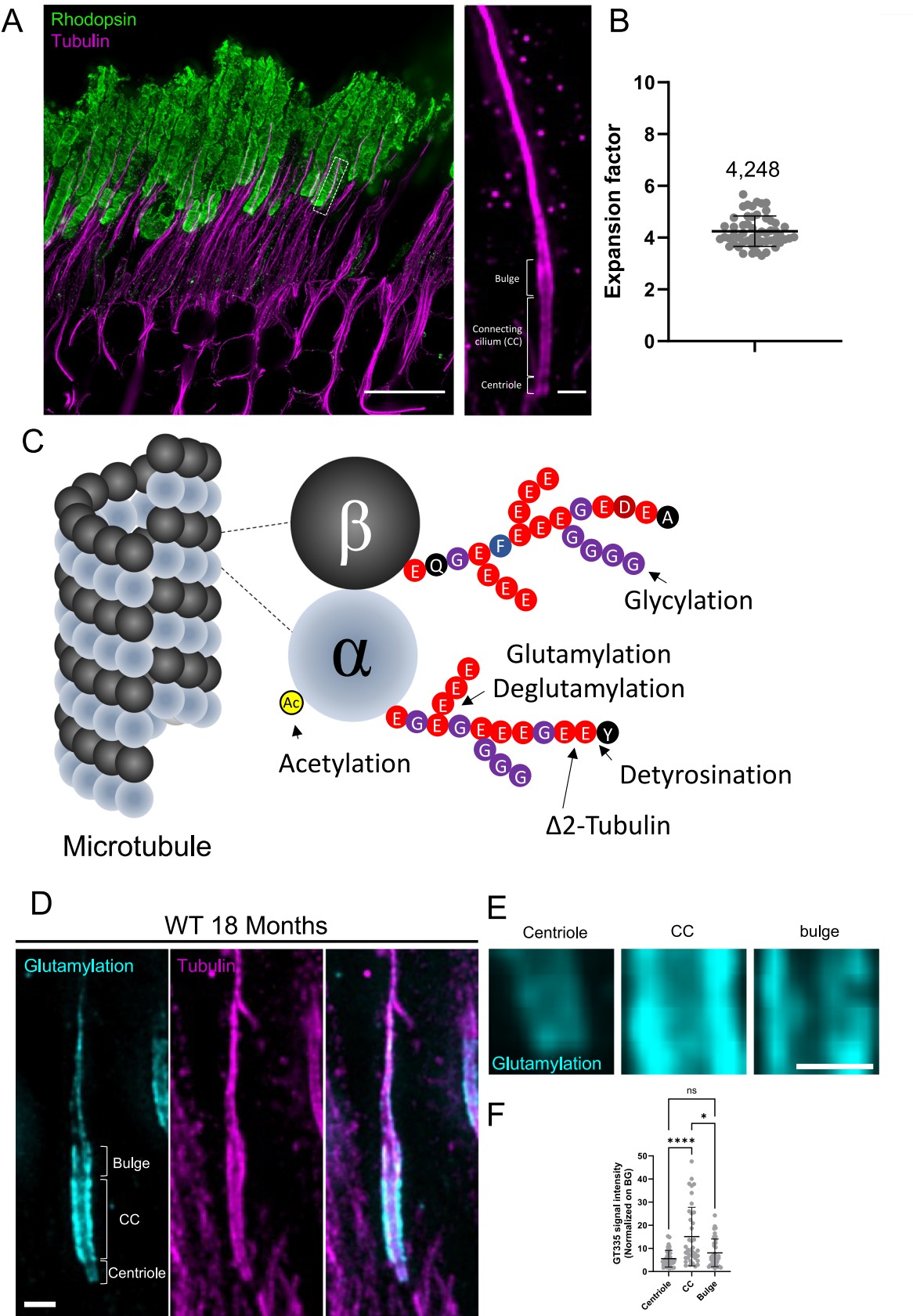

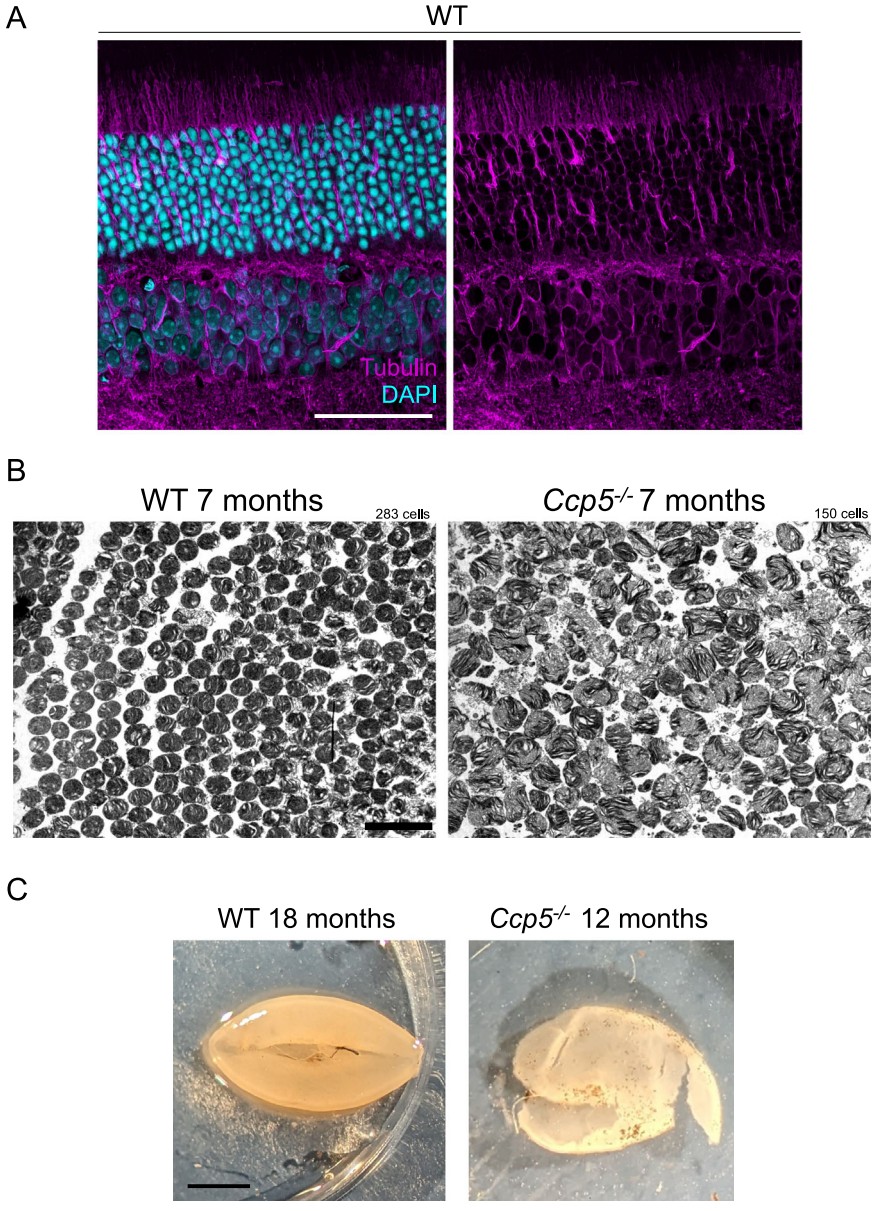

**Figure EV2. Global morphology of WT and *Ccp5*⁻/⁻ retinas.**

(A) Expanded WT mouse retina stained for tubulin (magenta) and DAPI (cyan). Note that ONL thickness can be measured only with tubulin staining, where nuclei position is clearly visible. Scale bar: 50 μm. (B) EM micrographs of 7-month-old WT or *Ccp5*⁻/⁻ retinas at low magnification to highlight defects in the organization of the membrane discs together with an important decrease in the number of cells (quantified on the top right). Scale bar: 5 μm. (C) 12-month-old WT (left) and *Ccp5*⁻/⁻ (right) retinas during dissection. Note that *Ccp5*⁻/⁻ retina is thinner and pigmented, reflecting a strong degeneration, a feature that we already observed previously (Faber et al, 2023). Scale bar: 1 mm.

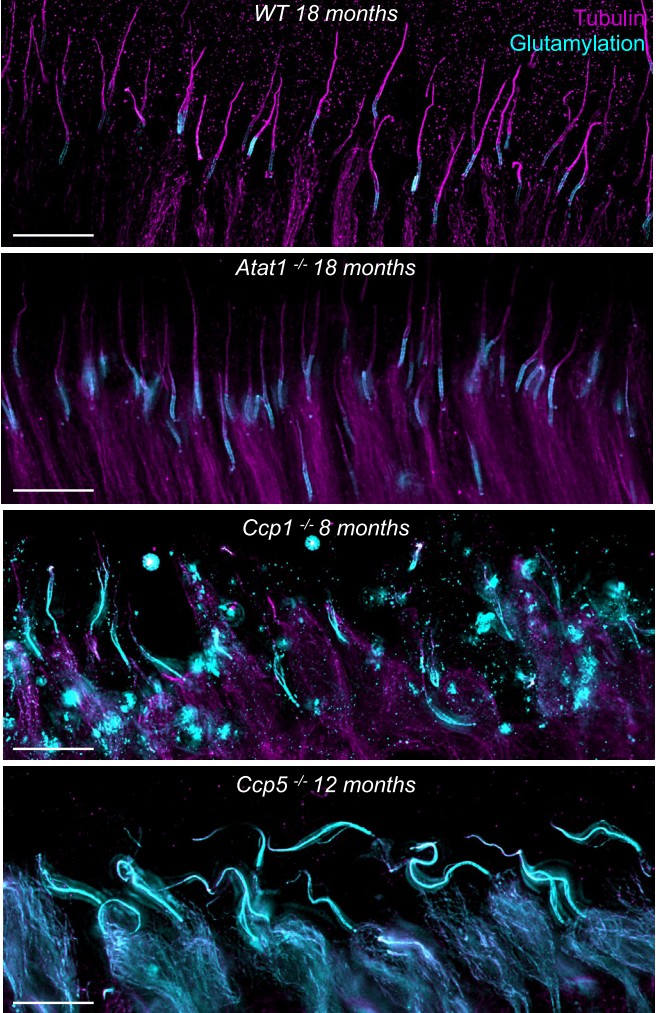

**Figure EV3.  Glutamylation level observed in several PTM mutants.**

Large field of view of WT, *Atat1⁻ᐟ⁻*, *Ccp1⁻ᐟ⁻* and *Ccp5⁻ᐟ⁻* expanded photoreceptor cell stained with GT335 (cyan) and tubulin (magenta). Note the intense glutamylation signal inside photoreceptor cell bodies in *Ccp5⁻ᐟ⁻* retina. Scale bar: 5 μm.

### WT 7 Months

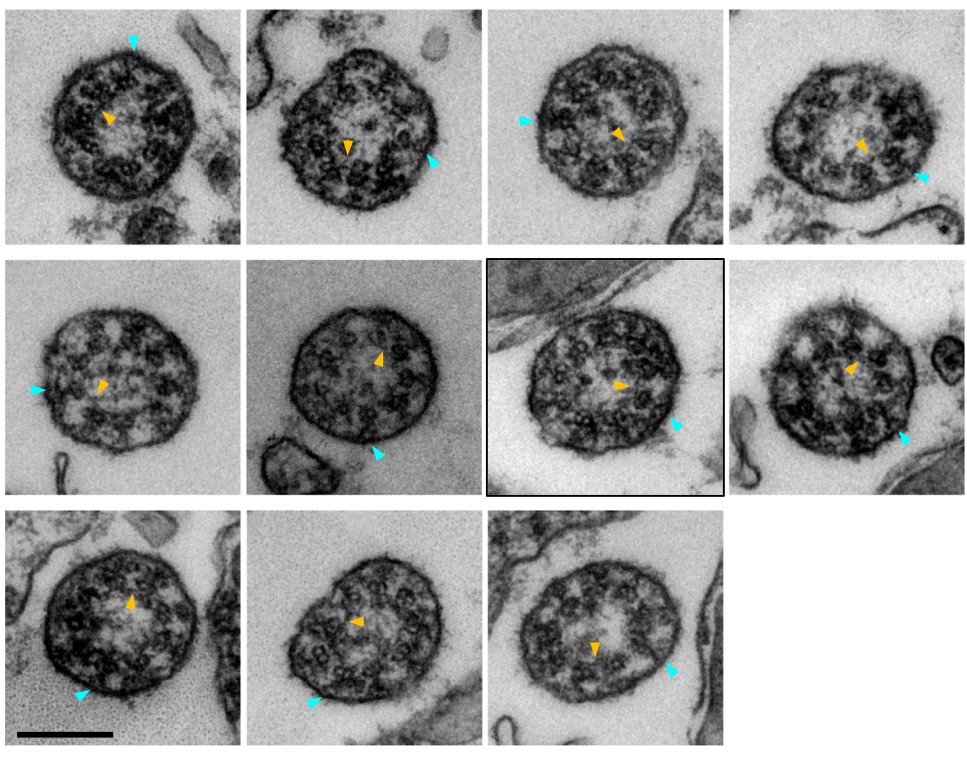

### Ccp5 <sup>-/-</sup> 7 Months

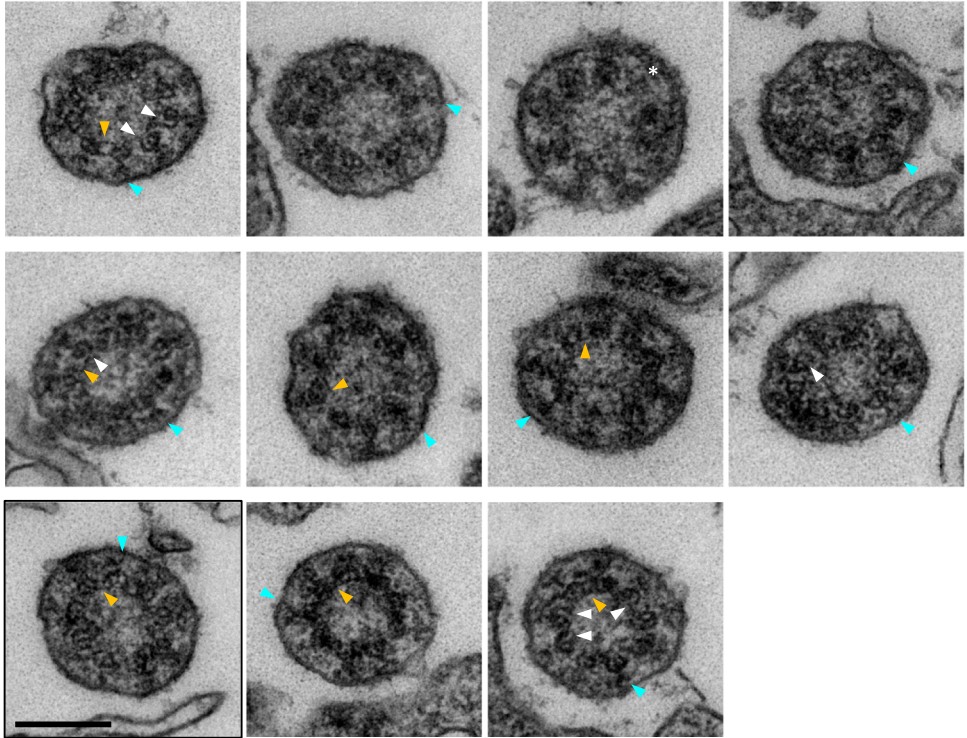

**Figure EV4. EM gallery of 7-month-old WT and *Ccp5*$^{-/-}$ photoreceptor CC.**

EM micrographs of 7-month-old WT and Ccp5−/− photoreceptor CC observed in transverse sections. White arrowheads highlight open B-tubules. Asterix depicts missing MTD. Cyan and orange arrowheads show Y-links and inner scaffold, respectively. The two images with black border are the ones used in the Fig. 6. Scale bar: 200 nm.

## Lca5$^{gt/gt}$ P18

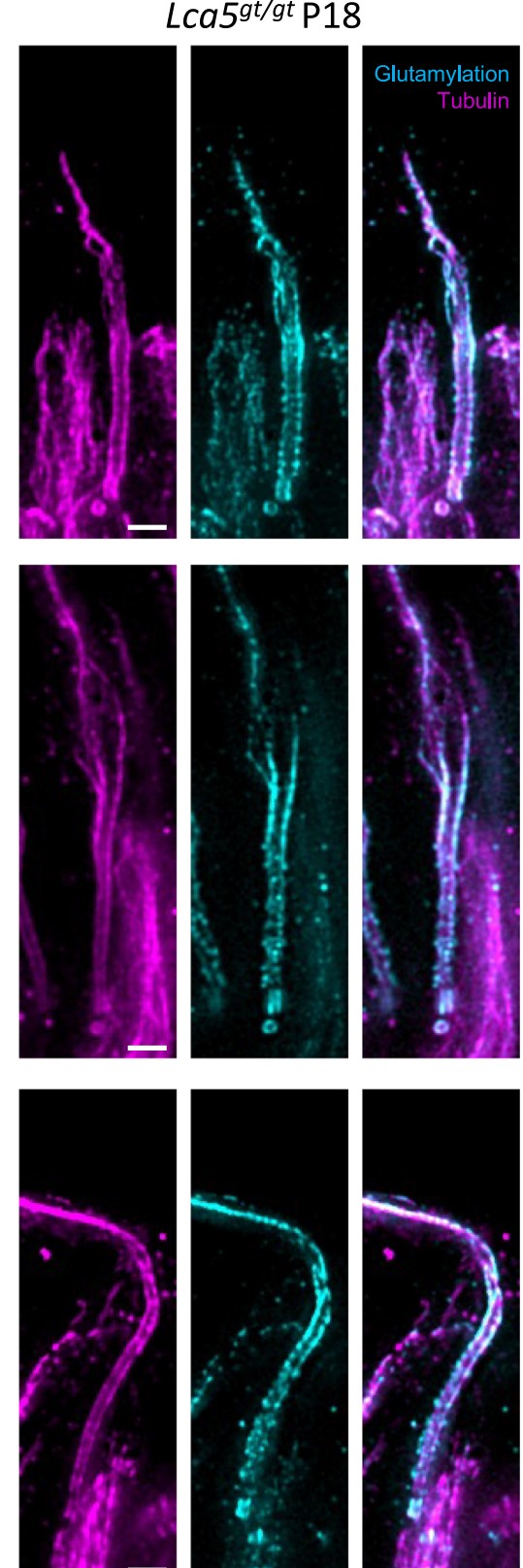

**Figure EV5.  Hyperglutamylation of the OS in _Lca5$^{gt/gt}$_ photoreceptor cells.**

Expanded P18 _Lca5$^{−/−}$_ photoreceptor cells stained for glutamylation (GT335, cyan) and tubulin (magenta). Scale bar: 500 nm.

