## [Peer Review File · The EMBO Journal]

Glutamylation imbalance impairs the molecular architecture of the photoreceptor cilium

Olivier Mercey, Sudarshan Gadadhar, Maria Magiera, Laura Lebrun, Corinne Kostic, Alexandre Moulin, Yvan Arsenijevic, Carsten Janke, Paul Guichard, and Virginie Hamel

Corresponding author(s): Virginie Hamel (virginie.hamel@unige.ch), Carsten Janke (Carsten.Janke@curie.fr), Paul Guichard (paul.guichard@unige.ch)

Review Timeline:

Transferred from Review Commons:	29th Jul 24
Editorial Decision:	1st Sep 24
Revision Received:	30th Sep 24
Editorial Decision:	14th Oct 24
Revision Received:	17th Oct 24
Accepted:	18th Oct 24

Editor: Ieva Gailite

Transaction Report:

This manuscript was transferred to The EMBO Journal following peer review at Review Commons.

Review #1**1. Evidence, reproducibility and clarity:****Evidence, reproducibility and clarity (Required)**

This is a manuscript that focuses on understanding the role of tubulin PTMs, especially glutamylation, during the development and function of photoreceptor cells. The investigators have generated CCP1^{-/-} and CCP5^{-/-} mice and investigated the effect of loss of CCP1/5 on tubulin PTMs and photoreceptor cilia formation. Overall it is a nicely written manuscript. All shown data are convincing, with very impressive images that are supported by quantifications. This is definitely strength of the manuscript. However, expansion microscopy does not provide sufficient resolution to allow going beyond descriptive study. For example, does the loss of glutamylation affect axonemal microtubule organization, which would lead to defects in photo-cilia formation and function? The only way to really answer this question is to do EM analysis of WT and CCP1 or CCP5 mutant cells. I do realize that doing EM analysis is not trivial, however, I think that EM analysis is absolutely crucial for this study since it would reveal what are the consequences of loss of CCP1 and CCP5 (leading to hyper-glutamylation) on ciliary microtubule organization, which is a main question of the manuscript. For other specific comments see below:

1. In all quantifications of tubulin PTMs are expressed and "length of the signal". While this is a valid measurement, it remains unclear whether the levels of glutamylation (intensity of glutamylation normalized to intensity of tubulin in the same area) in CCP1 or CCP5 mutant animals are affected in outer segment and connecting cilium. These analyses should be performed and shown in the manuscript. This would allow to determine whether CCP1/5 affects glutamylation in basal body, connecting cilium or outer segments (or all of them).
2. Is transition zone formation affected in these cilia? That should be tested and quantified.
3. What is that "sheath-like" signal present in both, anti-polyglutamylation and anti-detyrosination antibody-stained samples? Authors speculate that its polyglutamylation of other (non-tubulin) proteins. However, anti-detyrosination antibodies also produces the same signal, suggesting that it may be a non-specific signal. Do authors see the same "sheath-like" signal in CCP mutants?

****Referee Cross-commenting****

I do not dispute the need of these types of studies, even it is a bit descriptive. I still think EM would be extremely informative and would substantially increase the significant of the study. Having said that, I do not think the lack of EM precludes the publication of this study.

Simply it will affect its significance and the journal where the study will ultimately be published.

2. Significance:

Significance (Required)

While tubulin posttranslational modifications have emerged as very important aspect during regulation of tubulin dynamics, how these PTMs affect tissue development in vivo remains poorly understood. As the result, this study does have potential high significance and would be of interest to a scientific community. The strength of this study is the comprehensive analysis of tubulin PTMs in photoreceptor cells, as well as exploring the roles of two de-glutamylases CCP1 and CCP5 in the process in vivo. However, authors do not go beyond descriptive analysis of mice photoreceptors. As the result, the manuscript lacks any mechanistic insights. Thus, it remains unclear what is the function of glutamylation in photoreceptor cell development and function. Lack of this mechanistic information limits the significance of this study.

3. How much time do you estimate the authors will need to complete the suggested revisions:

Estimated time to Complete Revisions (Required)

(Decision Recommendation)

Between 3 and 6 months

4. Review Commons values the work of reviewers and encourages them to get credit for their work. Select 'Yes' below to register your reviewing activity at Web of Science Reviewer Recognition Service (formerly Publons); note that the content of your review will not be visible on Web of Science.

No

Review #2

1. Evidence, reproducibility and clarity:

Evidence, reproducibility and clarity (Required)

This study describes the molecular composition of the cilium of photoreceptor cells in wild type mice and mutants in which the deglutamylation enzymes Ccp5 and Ccp1 have been

knocked-out. These mutations induced the degeneration of photoreceptor cells. In particular, the study focuses on the spatial distribution of post-translational modifications of the tubulins forming the cilium and the associated morphological defects of the outer segments of the photoreceptors. The study is based on the use of expansion microscopy in order to accurately describe the organisation of microtubules at the distal part of the cilium and the radial distribution of tubulin PTM in mouse (and human !) retina. Beyond the mere but useful and original description of the patterns of PTM along the cilium, the comparison between WT and mutant cilia allowed the authors to propose how these modifications might contribute to photoreceptor cell degeneration.

****Major comments****

All conclusions are based on the analysis of immunostainings of the retina of WT and mutant mice that have been expanded by a protocol that had been developed and improved by the authors over the last few years. Samples were properly fixed, expanded, stained and imaged. Images have been well quantified so the conclusions are fully supported by the data. So this work does not require further experiments.

****Minor comments****

The patterns of tubulin PTM do not directly account for the cell degeneration and, as discussed by the authors, it is likely the defective transport along the cilium that is responsible for it. Hence the description of the localisation of IFT88 and LCA5 are important. I found surprising that IFT88 is absent from the centriole of Ccp5 and Ccp1 mutant although the centriole does not seem hyper or hypo glutamylated. Maybe it is. If so it would be worth being described and discussed. The functions of ccp5 and ccp1 could be more detailed in the introduction in order to allow the reader to appreciate the novelty of this study.

****Referees cross-commenting****

Response to Rev3:

I fully agree with this comment and would not ask for the EM.

2. Significance:

Significance (Required)

General assessment:

The tubulin code is a great hypothesis, according to which tubulin PTM encode some information along microtubules which regulate their function, and notably the type of cargo that are transported along them, but this hypothesis has not yet been demonstrated. The « code » is still under study and the consequence of this encoding are still unclear. To progress on these questions we need to find examples of physiological misfunctions that result from defective tubulin modifications. There are not so many examples and retina pigmentosa might be one. So the description of a tissue that is highly dependent on microtubules/cilia to function properly and which become defective when tubulin modifications are altered is a highly valuable system which might become a reference in the future to decode the tubulin code. A great outcome of this study is the description of multiple PTM patterns and their interdependence, ie the impact of hyperglutamylation on other modifications. Another important conclusion is the observation that the microtubule structure is disorganized in the distal part of cilia in mutant mice. While there are several examples of disruption of ciliary architecture upon the loss of structural proteins, I am not aware of many other examples of such consequence for tubulin modification enzymes.

Advance:

This study is « descriptive only », in the sense that it reports careful measurements of PTM patterns on different tissues without revealing how these modifications affect the transport through the cilium and induce the degenerescence of the outer segment of the photoreceptor cells. However such experiments might be difficult to perform in mice or in their retina. The advantage of this study is to provide a fine characterisation of tubulin PTM pattern in a tissue, but it is also its limitation since it is difficult to perform functional assay on tissues.

Audience:

This work will be of interest to a broad audience : to cell biologists interested on the tubulin code and cytoskeleton-based structures, to developmental biologists interested on ciliary functions or retina development and function, and to medical doctors interested in retina disorders and ciliopathies.

Reviewer expertise:

I am a cell biologist working on cell cytoskeleton and I am not aware of all recent progress on ciliopathies nor tubulin PTM.

3. How much time do you estimate the authors will need to complete the suggested revisions:

Estimated time to Complete Revisions (Required)

(Decision Recommendation)

Cannot tell / Not applicable

4. Review Commons values the work of reviewers and encourages them to get credit for their work. Select 'Yes' below to register your reviewing activity at Web of Science Reviewer Recognition Service (formerly Publons); note that the content of your review will not be visible on Web of Science.

No

Review #3

1. Evidence, reproducibility and clarity:

Evidence, reproducibility and clarity (Required)

This is well-executed and well-described study using recently developed expansion microscopy techniques to improved the resolution of our understanding of location-specific post-translational modifications of tubulin within mouse photoreceptor sensory cilia (outer segments). The paper also reports the effects on these modifications and overall retinal/photoreceptor morphology in mice with genetic deficiencies in deglutamylating (CC5- Agbl^{-/-} and CC1^{-/-}) or tubulin acyl-transferase (Ata1^{-/-}) enzymes.

The work is technically sound, but there are two suggestions. One is that the paper notes the possibility of a non-tubulin glutamyated protein within the connecting cilium. The authors may want to test with antibodies specific for RPGR, a protein known to be glutamylated to see if it co-localizes, using their methods, with the mystery cross-reactivity with the GT335 antibody. The other is that the authors reconsider their title. The basic retinal degeneration phenotype of the Agbl^{-/-} mice has previously been described. What makes this paper interesting is the new information on nanoscale localization of specific post-translational modifications and the ways that one PTM influences the others, and they

may want to capture these novel features in their title.

****Referee Cross-commenting****

My reading of the reviewers' comments is that there is consensus about the quality and significance of the work, but some differences over what additional results are needed to make a strong case for publication. In particular, the question of whether the authors should be required to obtain electron microscopy data is one that could have great impact on the time it would take to fulfill the reviewers' requests. There is no doubt that such data would be useful in addressing the structural details of the ciliopathy models studied in this MS, at a higher resolution even than is provided by the expansion/fluorescence microscopy. In this reviewer's opinion, such a requirement misses the real significance of the manuscript, which lies in the demonstration of the power of the imaging technology employed to provide new and informative details about interesting neurodegeneration models, rather than in the details themselves. This is not the first paper on the genotype in question and need not be the last, so other groups are free to pursue electron microscopy and other methods to delve even more deeply into the structural phenotype. Even then, it is a truism that no amount of structural detail can ever unambiguously reveal mechanism, so having such data would still not be the end of the story. To delay publication while the authors search for a way to collect the EM data risks having the field, in which technical advances are being made at a very rapid pace, pass by what is currently a paper at the cutting edge.

2. Significance:

Significance (Required)

The significance arises largely from reveal with unprecedented resolution, the locations within the photoreceptor sensory cilia of microtubules bearing certain post-translational modifications. The observation of retinal degeneration in knockouts of deglutamylating enzymes is not new, but is here revealed in much greater detail. Also of significance are the influence of glutamylation on glycylation and the lack of obvious effects of loss of tubulin acetylation.

3. How much time do you estimate the authors will need to complete the suggested revisions:

Estimated time to Complete Revisions (Required)

(Decision Recommendation)

Less than 1 month

Yes

Full Revision

Manuscript number: RC-2024-02462

Corresponding author(s): Virginie Hamel, Paul Guichard, Carsten Janke

1. General Statements [optional]

This section is optional. Insert here any general statements you wish to make about the goal of the study or about the reviews.

Reviewer #1 (Evidence, reproducibility and clarity (Required)):

This is a manuscript that focuses on understanding the role of tubulin PTMs, especially glutamylation, during the development and function of photoreceptor cells. The investigators have generated CCP1^{-/-} and CCP5^{-/-} mice and investigated the effect of loss of CCP1/5 on tubulin PTMs and photoreceptor cilia formation. Overall it is a nicely written manuscript. All shown data are convincing, with very impressive images that are supported by quantifications. This is definitely strength of the manuscript. However, expansion microscopy does not provide sufficient resolution to allow going beyond descriptive study. For example, does the loss of glutamylation affect axonemal microtubule organization, which would lead to defects in photocilia formation and function? The only way to really answer this question is to do EM analysis of WT and CCP1 or CCP5 mutant cells. I do realize that doing EM analysis is not trivial, however, I think that EM analysis is absolutely crucial for this study since it would reveal what are the consequences of loss of CCP1 and CCP5 (leading to hyper-glutamylation) on ciliary microtubule organization, which is a main question of the manuscript.

We thank the reviewer for this comment. We agree that expansion microscopy, even if it surpasses previous resolutions and provides unprecedented insights, is not sufficient to reveal subtle defects that could occur at the level of the microtubule ultrastructure in deglutamylase mutants (causing hyperglutamylation). However, thanks to expansion microscopy, we can now easily see in CCP mutants the overall structural defects at the level of axonemal microtubules. Indeed, we describe that the connecting cilium (CC) is mostly not impacted, even if exacerbated, whereas the distal cilium is highly affected, with open, curled or even broken microtubules. We also confirmed the integrity of the CC by staining two CC components (POC5 and CEP290), that have similar signals than in WT. We now changed the configuration of the **revised Figure 3** to better highlight the observed axonemal defects in the mutants.

We also undertook an EM analysis to fulfill the reviewer's request, aiming at providing a complementary analysis of the ultrastructure of CCP5 mutant photoreceptor axonemes. Thanks to this approach, we confirmed that the CC is mostly unaffected in CCP5 mutant, with the presence of both Y-links and the inner scaffold (new data can be found in revised **Figure 3J**,

Figure 6G and **Supplementary Figure 10**). We noticed sometimes open B-tubules, that may participate in IFT defects and OS collapse (data shown in **revised Supplementary Figure 10**). Finally, we demonstrate that the CC-associated Y-links and inner scaffold structure are found in the distal part of CCP5 mutant cilium, corroborating POC5 and CEP290 staining described using expansion microscopy (**revised Figure 6H, K, L**). We hope that these additions will convince the reviewer of the interest of this study beyond being descriptive.

For other specific comments see below:

1) In all quantifications of tubulin PTMs are expressed and "length of the signal". While this is a valid measurement, it remains unclear whether the levels of glutamylation (intensity of glutamylation normalized to intensity of tubulin in the same area) in CCP1 or CCP5 mutant animals are affected in outer segment and connecting cilium. These analyses should be performed and shown in the manuscript. This would allow to determine whether CCP1/5 affects glutamylation in basal body, connecting cilium or outer segments (or all of them).

Thank you for this comment. We agree that this information was missing. We have now provided a quantification of GT335 intensity in 3 different locations (centriole, connecting cilium (CC), and bulge in WT to highlight the intense signal observed at the CC (**revised Supplementary Figure 3A-C**). Furthermore, we compared the GT335 signal intensity in the distal cilium of WT and CCP5 mutant and confirmed the hyperglutamylation in this compartment in CCP5 mutant (**revised Figure 5J-L**). Of note, we decided to normalize the GT335 signal on background rather than on tubulin, in case the glutamylation level change has an impact on the tubulin signal itself.

2) Is transition zone formation affected in these cilia? That should be tested and quantified.

Thank you for raising this point. If by 'transition zone formation', the reviewer means the setup of the connecting cilium, we did not check properly at early time points of degeneration (between P4 and P30). However, in CCP1 and CCP5 mutants, the degeneration starts quite late (within few months), so we do not expect early events of ciliary formation to be impacted, that would lead to ciliary defects earlier. Furthermore, at late time points, we looked at two connecting cilium markers POC5 and CEP290, that reveal similar signal localization compared to WT, except that the CC length is increased (**revised Figure 6A, C, E, F**).

3) What is that "sheath-like" signal present in both, anti-polyglutamylation and anti-detyrosination antibody-stained samples? Authors speculate that its polyglutamylation of other (non-tubulin) proteins. However, anti-detyrosination antibodies also produces the same signal, suggesting that it may be a non-specific signal. Do authors see the same "sheath-like" signal in CCP mutants?

Full Revision

We thank the reviewer 1 for this comment. The presence of this sheath-like signal is really intriguing in WT, being too far from the tubulin to account for a tubulin modification. We now analyzed the GT335 signal evolution along outer segment (OS) early development (P10, P14, P22), and revealed that the sheath-like structure is not present at the very beginning of OS development (**revised Figure 1G-I**). Also, we compared the GT335 signal width in WT and CCP5 mutant CC and revealed that this external signal is lost in the mutant (**revised Figure 5J, K, M, N**). We hypothesized that the signal could come from another protein, such as RPGR, that has been shown to be glutamylated (Sun et al., 2016). To test this hypothesis, we stained for RPGR in WT and CCP5 mutants in mice, alongside a staining in human photoreceptor cells (**revised Figure 5O-S**). Of interest, we found that RPGR signal in human photoreceptors, which gave a better signal than in mice, is overlapping with the sleeve formed by GT335. This suggests that this RPGR could be part of this sheath-like structure. Furthermore, in CCP5 mutant, RPGR signal is lost, together with the GT335-associated sleeve, further hinting for a link between these two. RPGR being crucial for IFT and actin dynamics in the bulge, its loss could contribute to photoreceptor death. These new data and hypothesis have been integrated in the revised manuscript.

****Referee Cross-commenting****

I do not dispute the need of these types of studies, even it is a bit descriptive. I still think EM would be extremely informative and would substantially increase the significance of the study. Having said that, I do not think the lack of EM precludes the publication of this study. Simply it will affect its significance and the journal where the study will ultimately be published.

Reviewer #1 (Significance (Required)):

While tubulin posttranslational modifications have emerged as very important aspect during regulation of tubulin dynamics, how these PTMs affect tissue development in vivo remains poorly understood. As the result, this study does have potential high significance and would be of interest to a scientific community. The strength of this study is the comprehensive analysis of tubulin PTMs in photoreceptor cells, as well as exploring the roles of two de-glutamylases CCP1 and CCP5 in the process in vivo. However, authors do not go beyond descriptive analysis of mice photoreceptors. As the result, the manuscript lacks any mechanistic insights. Thus, it remains unclear what is the function of glutamylation in photoreceptor cell development and function. Lack of this mechanistic information limits the significance of this study.

We hope that with the additional experiments performed including EM analysis and RPGR staining, this study provides more mechanistic insights into the role of tubulin PTMs in photoreceptor cells.

Reviewer #2 (Evidence, reproducibility and clarity (Required)):

This study describes the molecular composition of the cilium of photoreceptor cells in wild type mice and mutants in which the deglutamylation enzymes Ccp5 and Ccp1 have been knocked-out. These mutations induced the degenerescence of photoreceptor cells. In particular, the study focuses on the spatial distribution of post-translational modifications of the tubulins forming the cilium and the associated morphological defects of the outer segments of the photoreceptors. The study is based on the use of expansion microscopy in order to accurately describe the organisation of microtubules at the distal part of the cilium and the radial distribution of tubulin PTM in mouse (and human !) retina. Beyond the mere but useful and original description of the patterns of PTM along the cilium, the comparison between WT and mutant cilia allowed the authors to propose how these modifications might contribute to photoreceptor cell degenerescence.

Major comments

All conclusions are based on the analysis of immunostainings of the retina of WT and mutant mice that have been expanded by a protocol that had been developed and improved by the authors over the last few years. Samples were properly fixed, expanded, stained and imaged. Images have been well quantified so the conclusions are fully supported by the data. So this work does not require further experiments.

We thank the reviewer for this supportive feedback.

Minor comments

The patterns of tubulin PTM do not directly account for the cell degenerescence and, as discussed by the authors, it is likely the defective transport along the cilium that is responsible for it. Hence the description of the localisation of IFT88 and LCA5 are important. I found surprising that IFT88 is absent from the centriole of Ccp5 and Ccp1 mutant although the centriole does not seem hyper or hypo glutamylated. Maybe it is. If so it would be worth being described and discussed.

Thank you for this comment. Indeed, the level of IFT proteins at the level of the ciliary entry is decreasing over time. We already described the same phenomenon in LCA5^{-/-} mice, where we hypothesized that LCA5 deficiency leads to the bulge region loss, thus impairing proper recycling of IFT components (Faber et al., 2023). In here, we think the same time course of events occurs, where loss of CCP5 somehow leads to the bulge loss, preventing the recycling of IFT that seems to be dispersed in the distal cilium (**revised Figure 4E, F**). Furthermore, we now show that RPGR is lost in CCP5 mutant, itself impairing IFT (**revised Figure 5O, S**). We have clarified this point in the revised manuscript.

Full Revision

The functions of ccp5 and ccp1 could be more detailed in the introduction in order to allow the reader to appreciate the novelty of this study.

Thank you for this comment. We have now included a paragraph in the introduction to better define CCP5 and CCP1 functions.

****Referees cross-commenting****

Response to Rev3:

I fully agree with this comment and would not ask for the EM.

Reviewer #2 (Significance (Required)):

General assessment:

The tubulin code is a great hypothesis, according to which tubulin PTM encode some information along microtubules which regulate their function, and notably the type of cargo that are transported along them, but this hypothesis has not yet been demonstrated. The « code » is still under study and the consequence of this encoding are still unclear. To progress on these questions we need to find examples of physiological misfunctions that result from defective tubulin modifications. There are not so many examples and retina pigmentosa might be one. So the description of a tissue that is highly dependent on microtubules/cilia to function properly and which become defective when tubulin modifications are altered is a highly valuable system which might become a reference in the future to decode the tubulin code. A great outcome of this study is the description of multiple PTM patterns and their interdependence, ie the impact of hyperglutamylation on other modifications. Another important conclusion is the observation that the microtubule structure is disorganized in the distal part of cilia in mutant mice. While there are several examples of disruption of ciliary architecture upon the loss of structural proteins, I am not aware of many other examples of such consequence for tubulin modification enzymes.

Advance:

This study is « descriptive only », in the sense that it reports careful measurements of PTM patterns on different tissues without revealing how these modifications affect the transport through the cilium and induce the degeneration of the outer segment of the photoreceptor cells. However such experiments might be difficult to perform in mice or in their retina. The advantage of this study is to provide a fine characterisation of tubulin PTM pattern in a tissue, but it is also its limitation since it is difficult to perform functional assay on tissues.

Audience:

This work will be of interest to a broad audience : to cell biologists interested on the tubulin code and cytoskeleton-based structures, to developmental biologists interested on ciliary functions

or retina development and function, and to medical doctors interested in retina disorders and ciliopathies.

Reviewer expertise:

I am a cell biologist working on cell cytoskeleton and I am not aware of all recent progress on ciliopathies nor tubulin PTM.

Reviewer #3 (Evidence, reproducibility and clarity (Required)):

This is well-executed and well-described study using recently developed expansion microscopy techniques to improved the resolution of our understanding of location-specific post-translational modifications of tubulin within mouse photoreceptor sensory cilia (outer segments). The paper also reports the effects on these modifications and overall retinal/photoreceptor morphology in mice with genetic deficiencies in deglutamylating (CC5- Agbl^{-/-} and CC1^{-/-}) or tubulin acyl-transferase (Ata1^{-/-}) enzymes.

****Referee Cross-commenting****

My reading of the reviewers' comments is that there is consensus about the quality and significance of the work, but some differences over what additional results are needed to make a strong case for publication. In particular, the question of whether the authors should be required to obtain electron microscopy data is one that could have great impact on the time it would take to fulfill the reviewers' requests. There is not doubt that such data would be useful in addressing the structural details of the ciliopathy models studied in this MS, at a higher resolution even than is provided by the expansion/fluorescence microscopy. In this reviewer's opinion, such a requirement misses the real significance of the manuscript, which lies in the demonstration of the power of the imaging technology employed to provide new and informative details about interesting neurodegeneration models, rather than in the details themselves. This is not the first paper on the genotype in question and need not be the last, so other groups are free to pursue electron microscopy and other methods to delve even more deeply into the structural phenotype. Even then, it is a truism that no amount of structural detail can ever unambiguously reveal mechanism, so having such data would still not be the end of the story. To delay publication while the authors search for a way to collect the EM data risks having the field, in which technical advances are being made at a very rapid pace, pass by what is currently a paper at the cutting edge.

The work is technically sound, but there are two suggestions. One is that the paper notes the possibility of a non-tubulin glutamylated protein within the connecting cilium. The authors may want to test with antibodies specific for RPGR, a protein known to be glutamylated to see if it co-localizes, using their methods, with the mystery cross-reactivity with the GT335 antibody.

Full Revision

The other is that the authors reconsider their title. The basic retinal degeneration phenotype of the *Agbl*^{-/-} mice has previously been described. What makes this paper interesting is the new information on nanoscale localization of specific post-translational modifications and the ways that one PTM influences the others, and they may want to capture these novel features in their title.

We thank the reviewer for these comments. We have now included a panel with RPGR staining in WT and *CCP5*^{-/-} mice, together with RPGR staining in human photoreceptor cells (**revised Figure 5O-S**). We show that RPGR staining coincides with the glutamylated sleeve observed with GT335 staining. Furthermore, in *CCP5* mutant, both glutamylated sheath-like structure and RPGR seem to be lost, corroborating the idea that RPGR could be part of this sleeve. Also, RPGR being associated with IFT and actin dynamics in the bulge, it is not surprising that its loss is accompanied by a collapse of the OS. We have now integrated these new data in the revised manuscript.

Furthermore, we have now changed the title:

“Glutamylation imbalance impairs the molecular architecture of the photoreceptor cilium”

Reviewer #3 (Significance (Required)):

The significance arises largely from reveal with unprecedented resolution, the locations within the photoreceptor sensory cilia of microtubules bearing certain post-translational modifications. The observation of retinal degeneration in knockouts of deglutamylating enzymes is not new, but is here revealed in much greater detail. Also of significance are the influence of glutamylation on glycylation and the lack of obvious effects of loss of tubulin acetylation.

Dear Dr. Hamel,

Thank you for submitting your revised Review Commons manuscript to The EMBO Journal. Your manuscript has now been seen by one of the original reviewers, who finds that their main concerns have been addressed and now recommend publication of the manuscript. I will therefore be happy to accept the manuscript for publication in The EMBO Journal after its reformatting along the guidelines included in the attached document.

Please feel free to contact me if you have any further questions regarding this final editorial revision. Please use the link below to upload the revised files.

Thank you for the opportunity to consider your work for publication, and I look forward to receiving your revised manuscript.

With best regards,

Ieva

Ieva Gailite, PhD
Senior Scientific Editor
The EMBO Journal
Meyerohofstrasse 1
D-69117 Heidelberg
Tel: +4962218891309
i.gailite@embojournal.org

We realize that it is difficult to revise to a specific deadline. In the interest of protecting the conceptual advance provided by the work, we recommend a revision within 3 months (30th Nov 2024). Please discuss the revision progress ahead of this time with the editor if you require more time to complete the revisions.

Referee #1:

Authors have addressed all my comments/concerns. In my opinions this revised manuscript is now ready for publication at EMBO Journal.

Response to reviewers

Referee #1:

Authors have addressed all my comments/concerns. In my opinions this revised manuscript is now ready for publication at EMBO Journal.

We thank the reviewer for his/her feedback

Note:

1- for Figure 1B. Looking for source data, we realized we had better top view images for Acetylated tubulin that we decided to replace (CC and centriole).

2- The quantification for the GT335 and tubulin signal width (From figure 1I) was not explained in the method. We now added a new paragraph.

3- Affiliation of the second author has been updated.

Dear Dr. Hamel,

Thank you for submitting a revised version of your manuscript and for addressing most of the remaining editorial points. I am afraid that there remain a few formatting aspects as outlined below that still need to be implemented in the manuscript:

1. Please include corresponding authors' email addresses on the front page.
2. Please provide up to five keywords.
3. Please move "Data Availability" section before "Acknowledgments".
4. Please rename "Declaration of competing interest" section into "Disclosure Statement and Competing Interests" and move after Acknowledgments.
5. Supplementary figure 7A is mentioned in the manuscript text, while there are only five Appendix figures. Please check and correct.
6. Table S1 is mentioned in the text, but is not provided - should the callout be updated to Reagents and Tools Table?
7. Please remove Reagents and Tools Table from the manuscript text and upload as a separate .docx file. choosing the file type "Reagent Table".
8. In our standard image integrity check, we noted that the same image has been reused in figure panels 6G and EV4. Please indicate this in the figure legend.
9. Our data editors have flagged the following issues in figure legends that need correcting:
 - Please provide the exact p values in the legends of figures 1i; 3e; 4g-l; 5e, h-i, n; 6c-d, f, j; EV 1f.
 - Please note that for the figure 2g, p-values and statistical tests are indicated in the legends. However, comparison for the same, ""*****"" has not been represented in the figure. Please rectify this in the figure or legend as applicable.
 - Please describe the nature of replicates (e.g., biological or technical) in the legend of figure EV 1b.
 - Please define the error bars in the legends of figures 4g-k.
 - Please provide the scale bar for figures 1b-e; 2b-e; 3b-d, f-i; 4b-f.
 - Please note that scale bar and its definition are missing for figures 1h; 5m; EV 1e.

I have also gone through the abstract and synopsis of the manuscript and would like to propose the edits included below and in the attached file. I have also written a short blurb that will accompany the title of your manuscript on our online page of contents. Please take a look and let me know if any corrections are needed.

Blurb:

Super-resolution ultrastructure expansion microscopy reveals the nanoscale localization of post-translational tubulin modifications in human and mouse photoreceptors.

Synopsis:

Tubulin post-translational modifications (PTMs) are highly enriched in the cilia and are important for photoreceptor architecture and function. This study reveals the nanoscale localization of tubulin PTMs along the cilium of mouse and human photoreceptor cells and implicates glutamylation in correct establishment of ciliary molecular architecture.

. Ultrastructure expansion microscopy reveals that most tubulin PTMs are enriched in the connecting cilium and bulge region of photoreceptor cilia.

. Hyperglutamylation upon loss of deglutamyases CCP1 or CCP5 leads to architectural defects of the axoneme and loss of the bulge region.

. Hyperglutamylation causes mislocalization or loss of ciliary components, including intraflagellar transport proteins.

. Hyperglutamylation is associated with extended connecting cilium.

Please feel free to contact me if have any questions regarding this final revision. Thank you again for giving us the chance to consider your manuscript for The EMBO Journal. I look forward to receiving the revised version.

With best regards,

Ieva

We realize that it is difficult to revise to a specific deadline. In the interest of protecting the conceptual advance provided by the work, we recommend a revision within 3 months (12th Jan 2025). Please discuss the revision progress ahead of this time with the editor if you require more time to complete the revisions.

The authors addressed the remaining editorial issues.

Dear Virginie,

Thank you for implementing the final edits and for your input on the textual changes. I am happy to go ahead with the corrected blurb. I am now pleased to inform you that your manuscript has been accepted for publication in the EMBO Journal.

If you have any questions, please do not hesitate to contact the Editorial Office. Thank you for your contribution to The EMBO Journal and congratulations on a nice study!

With best wishes,

Ieva

Rev_Com_number: RC-2024-02462
New_manu_number: EMBOJ-2024-118613R1
Corr_author: Hamel
Title: Glutamylaton imbalance impairs the molecular architecture of the photoreceptor cilium